# Freshwater Mussels, Ecosystem Services, and Clean Water Regulation in Minnesota: Formulating an Effective Conservation Strategy

Baishali Bakshi [1,*], R. William Bouchard, Jr. [1], Robert Dietz [1], Daniel Hornbach [2], Philip Monson [1], Bernard Sietman [3] and Dennis Wasley [1]

1 Minnesota Pollution Control Agency, Saint Paul, MN 55155, USA; will.bouchard@state.mn.us (R.W.B.J.); robert.dietz@state.mn.us (R.D.); phil.monson@state.mn.us (P.M.); dennis.wasley@state.mn.us (D.W.)
2 Department of Environmental Studies, Macalester College, Saint Paul, MN 55113, USA; hornbach@macalester.edu
3 Minnesota Department of Natural Resources, Lake City, MN 55041, USA; bernard.sietman@state.mn.us
* Correspondence: baishali.bakshi@state.mn.us; Tel.: +1-612-670-0152

**Abstract:** Freshwater mussels are threatened with extirpation in North America. They are a sentinel species for ecosystem function and contribute towards many ecosystem services. As mussels require clean water to survive, and since conserving ecosystem services is implicit in the federal Clean Water Act, incorporating mussel conservation into state water policies could serve multiple conservation goals. In this paper we conduct a comprehensive critical review of three topics related to freshwater mussels: their contribution to ecosystem services, their links with water quality, and threats to their survival from water pollutants and extent of protection available from regulation. In so doing, we identify gaps between the water quality requirements of mussels and the protection provided by current water regulation to help inform clean water and conservation goals in Minnesota. We find freshwater mussels to be generally sensitive to a wide variety of pollutants, and particularly to nutrients such as total nitrogen and total phosphorus and to major ions such as chloride. In addition, we find that current state water quality standards may not be sufficiently protective of mussels. We formulate a framework for determining an effective conservation strategy for mussels in Minnesota based on ecological and economic criteria to ensure adequate conservation at a reasonable cost.

**Keywords:** freshwater mussel conservation; ecosystem services; pollution; clean water regulation

## 1. Introduction

Freshwater mussels (Superfamily Unionoidea) (https://en.wikipedia.org/wiki/Unionidae (accessed on 5 July 2023)) have been declining at substantial rates (http://www.molluskconservation.org/MC_Ftpage.html (accessed on 5 July 2023)), with 30 North American taxa lost in the last century and 74% of remaining species currently threatened with extirpation [1–4]. North America being the richest biodiversity hotspot for freshwater mussels with ~300 species recorded, makes their imperilment of global concern [5].

Anthropogenic impacts are the main cause of mussel decline, within which overharvesting, impoundments, and other hydrological changes have been historically important. Overharvesting is documented as a historical cause of population decline due to the pearl button industry and remains a threat owing to the cultured pearl industry [6,7]. In recent decades many other threats have emerged, including invasive species [8,9], disease [10], increased nutrient and sediment loadings from agricultural and commercial activities [11], toxic pollution [12], and climate change [13]. For example, research has attributed the significant freshwater mussel declines observed over the past 35 years in the biodiverse Clinch River in the southeastern United States to water pollution from ammonia ($NH_3$),

heavy metals, and major ions due to wastewater treatment plants (WWTPs), mining, and agriculture, as opposed to modification of the physical habitat, as previously believed [14].

In Minnesota, fifty-one species of freshwater mussels have been recorded, with 8% lost and 60% of remaining species under threat from a variety of factors, including water pollution from nutrients, sediments [15], major ions [16,17], mining operations, intensive agricultural land use 87-, climate change [18,19], genetic factors [20], and even unknown factors, which highlight the need to account for uncertainty [21].

Freshwater mussels are building blocks of aquatic ecosystems, contributing substantially towards valuable benefits from nature called ecosystem services, such as food, water filtration, water quality, nutrient cycling, wildlife habitat, and cultural services; moreover, they play an important role in the food web [22]. A substantial part of the world's ecosystems and biodiversity have been lost due to human activity in the last fifty years and an additional 60% are being degraded and/or unsustainably used [23] (https://www.millenniumassessment.org/en/Condition.html (accessed on 5 July 2023)). The 2019 global assessment report by the Intergovernmental Science-Policy Platform on Biodiversity and Ecosystem Services (IPBES) found that human drivers have significantly altered 75% of global land surface, leading to increasing cumulative impacts for 66% of oceans and to the loss of 85% of wetlands. These alterations in natural ecosystems have serious consequences. For example, freshwater ecosystems face multiple threats due to land-use changes, mainly agricultural expansion, and about 1 million global species are threatened with extinction unless effective action is taken to restrict the drivers of biodiversity loss [24].

Effective conservation of ecosystem services is necessary to protect natural ecosystems while sustaining their continued use for humans and other species in the presence of global changes in population, urbanization, agriculture, climate, and natural habitats. Consequently, ecosystem service-oriented conservation approaches are being increasingly considered and adopted in public policy [25–34]. As ecosystem services are typically non-marketable, it is difficult to estimate changes in their flow due to any given stressor. Insufficient knowledge about the structure, functions, and linked networks of the underlying ecosystems as well as spatiotemporal impacts from global change further complicate the estimation process by adding multiple measures of uncertainty. The role of ecosystem services in human well-being, combined with their complex valuation process, demonstrates the importance of their consideration in environmental decision-making. Consequently, effective conservation of declining species such as freshwater mussels that contribute substantially towards provisioning of ecosystem services has value from a species perspective as well as from a more general need to sustain the flow of ecosystem services and human well-being for current and future generations.

Freshwater mussels are among the most sensitive species to water pollution; hence, they are key water quality indicators, meaning that policies for mussel conservation are in essence those that preserve and improve the quality of public water resources and sustain the ecosystem services they provide for human well-being and aquatic life uses. As mussels require clean water to survive, and since conserving ecosystem services is an implicit goal of the Federal Clean Water Act (Section 101.(a) of the Clean Water Act states: "The objective of this Act is to restore and maintain the chemical, physical, and biological integrity of the Nation's waters," while subpart (2) refers to the preservation of several ecosystem services: "It is the national goal that wherever attainable, an interim goal of water quality which provides for the protection and propagation of fish, shellfish, and wildlife and provides for recreation in and on the water be achieved..." https://www.epa.gov/sites/production/files/2017-08/documents/federal-water-pollution-control-act-508full.pdf (accessed on 5 July 2023)). Thus, incorporating mussel conservation into clean water goals could be a cost-effective strategy serving multiple conservation goals.

For Minnesota, a state facing substantial impacts from climate change due to its location in the boreal biome, which also contains one-third of the United States' freshwater resources and 17% of its remaining freshwater mussel species, the role of state water quality

protection can go a long way towards conserving mussels and other water resources for the future. Yet, we find that current water quality regulations in Minnesota are not sufficiently protective of freshwater mussels.

As loading of nutrients and of salts such as chlorides, sulfates, metals, and chemicals of emerging concern are a key threat to water quality in Minnesota (https://www.pca.state.mn.us/water/nutrient-reduction-strategy (accessed on 5 July 2023)) [35], effective freshwater mussel conservation depends on a comprehensive understanding of the links between mussels and ecosystem services, pollutants and their sources, [9], and mussel response to water pollution and current state standards, while accounting for the costs and benefits of regulating water quality.

The continued imperiled status of freshwater mussels and their links with water quality and ecosystem services raises the need to develop and implement an effective conservation strategy to serve existing water quality goals, build resilience against the future, and sustain ecosystem services for human well-being. In this paper, we present freshwater mussel conservation as an important problem for which a solution serving multiple objectives, including sustaining ecosystem services and clean water goals, is needed. We conduct a comprehensive review and critical analysis on the status of freshwater mussels and their linkages with water quality and pollutants in order to identify gaps between their water quality needs and existing water protections, providing helpful information to decision-makers. Our final result is a framework for an effective conservation strategy for mussels in Minnesota based on ecological and economic criteria.

## 2. Materials and Methods

The methods we employ are a combination of: (1) a literature review on three topics, namely, the *ecosystem services* literature showing the contribution of freshwater mussels, the *toxicological* literature documenting the threats faced by mussels from water pollution, and the *water quality regulation* literature showing the role of regulation in reducing environmental degradation; and (2) a critical analysis of past findings and current water quality standards.

## 3. Literature Review and Analysis in Three Topics

In order to understand the role of water quality regulation in mussel conservation and the development of an effective conservation strategy using regulation, we need to study the linkages between mussels and three different topics: the role of mussels in providing valuable ecosystem services, the relationship of mussels with water quality, and the threats to mussels due to water pollution and the role of clean water regulation in pollution reduction. We provide a critical analysis of each topic in this section. To the best of our knowledge, this is the first review of combined and disparate literature that collectively provides information on both conservation and clean water goals.

### 3.1. Mussels and Ecosystem Services

Freshwater mussels play a key role in aquatic ecosystems, providing valuable ecosystem services comparable to beneficial uses, which "Identify how people, aquatic communities, and wildlife use our waters" (https://www.pca.state.mn.us/business-with-us/water-quality-standards (accessed on 5 July 2023)). Ecosystem services include provisioning services, such as food and fuel; regulating services, such as climate regulation and pollination; supporting services, such as soil formation and nutrient cycling; and cultural services, such as recreation and spiritual well-being [36]. Freshwater mussels affect aquatic ecosystems through their relationship with phytoplankton, which are responsible for up to 85% of primary production in the world (https://earthsky.org/earth/how-much-do-oceans-add-to-worlds-oxygen/ (accessed on 5 July 2023), https://www.nationalgeographic.com/science/article/source-of-half-earth-s-oxygen-gets-little-credit (accessed on 5 July 2023)) [37]. Freshwater mussels feed on phytoplankton and convert the nutrients into their own tissue, biodeposits that settle on riverbeds, and to dissolved nutrients, that are

more easily cycled and accessible to other species [13]. Mussels provide ecosystem services belonging to all these categories, including nutrient cycling and storage, water purification through biofiltration and bioaccumulation, structural habitat, substrate, and food web modification, and water clarity [22].

### 3.1.1. Food Provisioning, Aquatic Recreation, and Cultural Services

Freshwater mussels are an important food source for several animal species [38–40] as well as for humans [41–45]. Their role in the food web helps to sustain aquatic recreation, including fishing [22]. Freshwater mussels provide cultural services such as material for tools/jewelry (http://www.iucnredlist.org/details/166247/0 (accessed on 5 July 2023)) and spiritual enhancement [22]. They have been culturally important to Native American tribes, both historically, and in modern times [5,46,47].

### 3.1.2. Habitat Provisioning

Mussels are 'ecosystem engineers' that help to build healthy species-rich communities across spatial scales by making habitats more suitable [48,49]. Unionid mussels often live in multispecies assemblages called mussel beds and in dense patches within these mussel beds [50]. An analysis across these multiple spatial scales in U.S. rivers found that the total densities of macroinvertebrates and dominant groups were significantly higher in patches containing mussels than where mussels were absent, with densities of macroinvertebrates being positively correlated with unionid density [49]. Mussel assemblages explained almost half of the variation in macroinvertebrate assemblages at both spatial scales, even after removing effects of similar habitat (environmental variables) and biogeographic history (spatial variables), documenting their importance in habitat preservation for macroinvertebrates. Mussels have been documented to have symbiotic associations with certain species of non-biting midges (family Chironomidae) [51,52], which are themselves indicator species for good water quality [53].

### 3.1.3. Biofiltration, Nutrient Cycling, and Nutrient Storage

Freshwater mussels' biofiltration ability has been documented to be remarkable, though filtration rates could depend on other factors such as physiology, abundance, species composition, and environmental conditions such as water quality and food availability. Mussels filtered 13.5% of the total phosphorus (TP) load of Lake St. Clair (a freshwater lake within the Great Lakes system, located between Michigan, USA and Ontario, Canada: https://en.wikipedia.org/wiki/Lake_St._Clair (accessed on 5 July 2023)) and provided >60% of this TP load as food to macrophytes and deposit feeders through biodeposition [54]. Mussels filtered the margins of a lake in Bangladesh in 21 h despite high nutrient pollution, providing both biofiltration and water clarity [55]. The filtration capabilities of duck mussels (*Anodonta anatina*) have been documented at 2.75 L/h per mussel [56]. Sustainable aquaculture of duck mussels has been considered for reducing nutrients and improving surface water quality and regional tourism in the Szczecin Lagoon, a coastal system in the Baltic Sea [56], and more generally for reservoirs [57]. A specific example of the significant filtration capacities of freshwater mussels in North America comes from a 480 km (298 mile) reach of the Upper Mississippi River (UMR), where mussels reportedly filtered ~53 million m$^3$/day (14 billion gallons) of water, as compared to only 0.7 million m$^3$/day (0.19 billion gallons) filtered by the Metropolitan Wastewater Treatment Plant (Metro Plant) (https://metrocouncil.org/Wastewater-Water/Publications-And-Resources/WASTEWATER/TREATMENT-PLANTS/Metropolitan-Wastewater-Treatment-Plant.aspx (accessed on 5 July 2023)) in Saint Paul, which is responsible for processing waste for the seven-county metropolitan area [58].

High filtration capabilities combined with long life spans enable mussels to bioaccumulate and either biodeposit or retain pollutants, including metals and nutrients, suggesting an important role as a biomarker as well as a mitigator with respect to water pollution [59]. Water pollution from heavy metals or metallic elements (MEs) is mainly attributed to

human activity [60]. MEs are highly toxic to many species, including humans, at very low concentrations [61], and removal to safe levels is currently beyond the capability of most WWTPs [62]. Freshwater mussels have been found to bioaccumulate and to remove metals, including cadmium, copper, lead, mercury, and zinc, from water at significant rates [63–67].

Freshwater mussels such as *Anodonta californiensis* and *Corbicula fluminea* (Asian Clam) have been found to efficiently remove emerging chemicals such as PFAS (pharmaceuticals, personal care products, pesticides, and flame retardants) as well as pathogens such as *E. coli* and the avian influenza virus from water [68–70]. The Zebra Mussel (*Dreissena polymorpha*) and Asian Clam have been reported to remove nutrient loads, reduce bacterial growth, and improve water quality in aquaculture [71,72], as an alternative to traditional wastewater treatment [62,73], and in specific industrial settings such as wineries [74].

However, bioaccumulation of contaminants in mussels and the resulting impacts of those contaminants are detrimental to mussel survival [75,76], which is noteworthy considering the recent widespread and continued decline in freshwater mussel species [1,2].

### 3.1.4. Valuation of Mussel-Based Ecosystem Services

In general, non-market valuation methods are appropriate for valuing ecosystem services provided by mussels, either based on willingness-to-pay (WTP) surveys [77] or on the value of similar market-based services such as water filtration infrastructure, wastewater treatment costs, replacement costs of treating sewage, and nutrient credit programs for evaluating mussel-based ecosystem services [78,79].

The American Fisheries Society has recommended using replacement cost as a measure of restitution for killed mussels [80]. Replacement costs include the costs of producing an equivalent number of juvenile mussels, restocking the kill site, and the costs of investigating, monitoring, and administering the mussel kill incident. Guidelines for estimating replacement cost and dollar values by mussel species are provided as a reference for assigning damages for mussel kills not involving threatened or endangered species. The replacement cost for species in Minnesota reported for 2002 and unadjusted for inflation was estimated to be USD 0.44–9.63 per juvenile mussel, depending on the species. However, species and their habitats have both use and non-use values for humans, as well as ecological values [81,82]. Through the provision of food and habitat for other species, freshwater mussels indirectly provide ecosystem benefits to humans, including recreational, spiritual, and cultural benefits [22]. Replacement costs, such as the estimates for killed mussels above, do not measure use, non-use, or ecological values [83], and depending on the region, species, and usage, these values may be substantial.

Three ecosystem services, namely, biofiltration, nutrient recycling (nitrogen, N and P), and nutrient storage (N, P, and carbon, C), were estimated using laboratory-derived physiological rates and river-wide estimates of species-specific mussel biomass in the Kiamichi River in southeastern Oklahoma from 1992 to 2011 [59]. These ecosystem services declined by 60% over the 20-year research period, similar to the decline in mussel abundance in the river. A survey of stakeholders in the Kiamichi study found that habitat for species and water quality have the highest economic value to people, with WTP estimates of USD 20.34 and 9.59, respectively, on an annual per household basis [77,84].

Due to the complex links between mussels and their ecosystems and the limited knowledge and data on the final ecosystem services they produce, robust dollar estimates of the ecosystem services provided by freshwater mussels are not yet available. However, the ecosystem goods and services provided by shellfish aquaculture towards reducing eutrophication in the coastal waters of the European Union have been estimated at USD 26–30 billion per year [85].

### 3.2. Mussels and Water Quality

Freshwater mussels are among the most sensitive aquatic species, being significantly affected by a wide range of pollutants, including contaminants of emerging concern (CECs) [86–88], major ions such as chloride and sulfate [89,90], metals [66,91], and exces-

sive loading of nutrients such as total phosphorus (TP) and total nitrogen (TN) [92,93]. TP comprises all forms of phosphorus, including organic phosphates found in plant or animal tissue and soluble inorganic forms such as orthophosphates taken up by plants and typically found in wastewater (http://bcn.boulder.co.us/basin/data/NEW/info/TP.html#:~:text=Total%20phosphorus%20[TP]%20is%20a,taken%20up%20by%20plant%20cells (accessed on 5 July 2023)). Similarly, TN includes nitrogen from both organic and inorganic sources [94] (https://www.falmouthma.gov/DocumentCenter/View/1129/Understanding-the-Basic-Principles-of-Nitrogenby-Robert-Scott-PDF?bidId=#:~:text=Total%20Nitrogen%20[TN]%20is%20the,measure%20nitrogen%20at%20wastewater%20plants (accessed on 5 July 2023)). The organic part comprises various compounds in living matter, such as amino acids and urea, of the general formula R-NH$_2$. The mineralization of these compounds by microorganisms produces inorganic nitrogen in the form of ammonia (NH$_3$) or ammonium (NH$_4$) [95,96] (https://www.sciencedirect.com/topics/earth-and-planetary-sciences/ammonification (accessed on 5 July 2023)). The aggregation of organic nitrogen, ammonia, and ammonium is called Total Kjeldahl Nitrogen (TKN) (https://en.wikipedia.org/wiki/Kjeldahl_method (accessed on 5 July 2023)). The remaining inorganic parts comprise nitrogen in nitrite (NO$_2$) and nitrate (NO$_3$) compounds [94].

Nutrient pollution can lead to eutrophic conditions in waterbodies due to excessive algae growth and oxygen depletion, adversely affecting aquatic life and recreation in many ways, from impacts on sensitive species such as freshwater mussels to impeding navigation and aquatic recreation such as fishing and swimming. The sources of nutrient pollution are mainly anthropogenic, including point and non-point sources such as municipal wastewater, industrial waste, and agricultural runoff. Currently, 40% of Minnesota waterbodies are impaired for the beneficial uses of aquatic life and recreation [97] (https://www.pca.state.mn.us/water/minnesotas-impaired-waters-list (accessed on 5 July 2023)). Based on the United States Environmental Protection Agency's (USEPA) 2013–2014 National Rivers and Streams Assessment, 58% (706,754 miles) of rivers and streams were rated 'poor' for phosphorus pollution relative to a least-disturbed reference distribution, while 43% (522,796 miles) were rated poor for nitrogen pollution (https://www.epa.gov/national-quatic-resource-surveys/national-rivers-and-streams-assessment-2013-14-key-findings (accessed on 5 July 2023)), showing that nutrient pollution is a significant water quality problem on a national scale. Therefore, it is important to reduce nutrient pollution in order to protect sensitive species such as freshwater mussels, to ensure that the state's waters are fishable and swimmable, and to reduce the spread of nutrient pollution in the Gulf of Mexico. Minnesota has adopted a Nutrient Reduction Strategy (NRS) outlining long-term reduction targets for TN and TP in order to meet these broad goals [35] (https://www.pca.state.mn.us/water/nutrient-reduction-strategy (accessed on 5 July 2023)).

Mussels are long-lived organisms (http://www.biologicaldiversity.org/campaigns/freshwater_mussels/ (accessed on 5 July 2023)), with certain species capable of living up to 100 years [98]. They grow slowly and are sensitive to disturbances such as changes in the climate, flow regime (http://www.water.wa.gov.au/water-topics/waterways/threats-to-our-waterways/altered-flow-regimes (accessed on 5 July 2023)), and land use, meaning that the impact of a given environmental stressor might only be observed after a long time has elapsed. This highlights the importance of accounting for uncertainty in mussel conservation. Freshwater mussels develop annual rings in their shells that can provide information on their growth rate and age [99].

As filter feeders on primary producers such as algae, mussel shells can reflect the effects of long-term human impacts on primary productivity, and consequently on environmental conditions and food sources in aquatic ecosystems [100]. A study in the Illinois River used sclerochronology (the study of physical and chemical variations in the accretionary hard tissues of invertebrates and coralline red algae and the temporal context in which they formed, particularly useful in the study of marine paleoclimatology: https://en.wikipedia.org/wiki/Sclerochronology (accessed on 5 July 2023)) and stable isotope analyses from shells of two species of freshwater mussels (*Amblema plicata* and *Quadrula quadrula*), using

specimens collected between circa 850 and 2013 in order to examine changes over historical time in age, growth, and food sources [101]. Mussel size and growth rates increased over the past 1000 years, especially after 1900, with individuals circa 2013 being double the size of those from 850. This post-1900 increase in growth rates was accompanied by a drastic reduction in mussel abundance and diversity, which the authors associated with widespread declines in dissolved oxygen (DO) levels; which only improved after 1966. In addition, the authors found increases in the isotopic signatures of both carbon and nitrogen in mussel shells after 1900. However, while carbon signatures stabilized to historical levels after the 1960s, nitrogen concentrations remained at elevated levels. These findings reflect the effect of marked increases in the early part of the 20th century in impoundments, nutrient pollution from various sources (including untreated wastewater), and consequent eutrophication, followed by water quality improvements in more recent times, which caused ecosystem-level changes in the river, affecting the mussels' environment and food sources. In particular, the reduction in carbon concentration in mussel shells is likely due to improved sewage treatment required by water quality legislations such as the Clean Water Act, while the continued high concentrations of nitrogen indicate that nutrient pollution from anthropogenic sources such as fertilizers, animal operations, and urban wastewater [102] remains a significant threat to water quality and sensitive species. Climate change and accompanying low-flow regimes in the future are likely to add to the existing pressures on mussel populations.

Recent studies in the Great Lakes region show that water pollution from point and non-point sources along with habitat modification from land use changes, particularly agriculture, are contributing factors in the decline of mussels. For example, a recent study in the Grand River in Ontario, Canada, a part of the Great Lakes watershed, revealed a four-mile dead zone (98% reduction in abundance) downstream of a WWTP in a location where historically mussels had been abundant [103]. The water near the plant had very high levels of $NH_3$ and very low levels of DO, while other nearby downstream segments of the river had high $NH_3$ as well as high nitrates ($NO_3^-$), phosphorus (P), chloride (Cl), and metals (http://greatlakesecho.org/2017/05/01/the-dead-zone-how-our-wastewater-is-killing-mussels/ (accessed on 5 July 2023)). The same study found a 60% decline in mussel abundance downstream of the urban area served by the WWTP compared to upstream areas. Additionally, mussels found downstream featured larger size classes, indicating population effects on smaller mussels.

Fifty-one species of freshwater mussels have been recorded in Minnesota, with 8% extirpated and 60% of the remaining species currently listed as endangered, threatened, or of special concern (See page 4: https://files.dnr.state.mn.us/natural_resources/ets/endlist.pdf (accessed on 5 July 2023)), making the state generally representative of the status of freshwater mussels in North America (Minnesota Department of Natural Resources (MDNR)).

Figure 1 shows mussel extirpation and abundance in all eighty-one Minnesota watersheds according to survey-based data from the MDNR. The extirpation rate is based on the ratio of shells of dead mussel species relative to the total number of species found, while abundance is denoted by the number of live mussels collected per minute.

Based on Figure 1, higher extirpation rates and lower abundance have been associated with agricultural regions as well as regions with higher human density in southern Minnesota, suggesting both point and non-point sources as possible causes of mussel decline. Mussel studies conducted by the MDNR during 1999–2012 found that only 23 of 40 species historically present in the largely agricultural Minnesota River basin (MRB) remained extant (https://danhornbach.org/wp-content/uploads/2022/11/MNRiverMusselsTable.pdf (accessed on 5 July 2023)). These studies found a significantly larger distribution of mussels in the northern part of the basin compared to the southern part, which is plausible as the latter has higher human density and environmental disturbance (https://danhornbach.org/wp-content/uploads/2022/10/MusselsHUC8.pdf (accessed on 5 July 2023)). Recent studies have confirmed further declines in mussel species in the MRB (https://danhornbach.org/wp-content/uploads/2022/10/MusselAbundanceMean

CPUEComparison.pdf (accessed on 5 July 2023)), including differential effects based on life history traits (the loss of equilibrium species and gain in periodic and opportunistic species) [5,104]. Equilibrium species have high juvenile survival, low numbers of offspring, and late reproductive maturity; periodic species have low juvenile survival, high numbers of offspring, and late maturity; and opportunistic species have low juvenile survival, low numbers of offspring, and early maturity. These results on impacts on species classified according to life history traits, were based on increasing levels of disturbance (https://danhornbach.org/wp-content/uploads/2022/10/LifeHistoryTraits.pdf (accessed on 5 July 2023)), drawing attention to sediment and nutrient pollution as well as habitat alteration from agricultural land use as possible contributing factors to these impacts [18,19]. As mussels are dependent on fish hosts to complete their life cycles, factors that affect host fish species can also affect mussels.

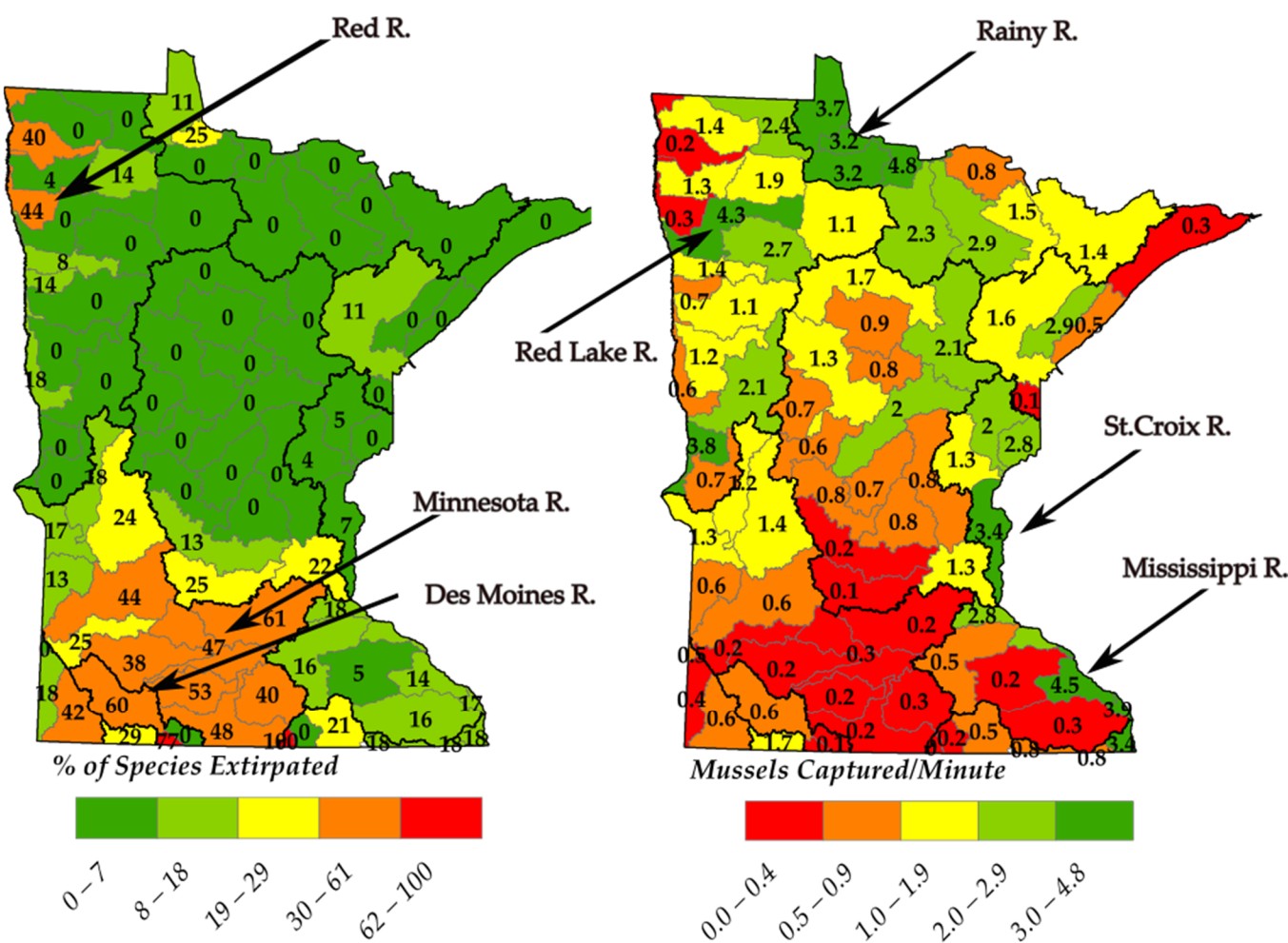

**Figure 1.** Mussel extirpation and abundance by watershed in Minnesota.

Figure 2 illustrates the current drivers of change for freshwater mussels. In addition, freshwater mussels and their conservation face challenges from exogenous drivers such as climate change, which could affect land and water use and underlying ecosystems in the near future. Being long-lived and highly sensitive species, their interaction with current and future threats is harder to study and predict, implying disparities in available data and understanding, as well as disparities between science and conservation policies.

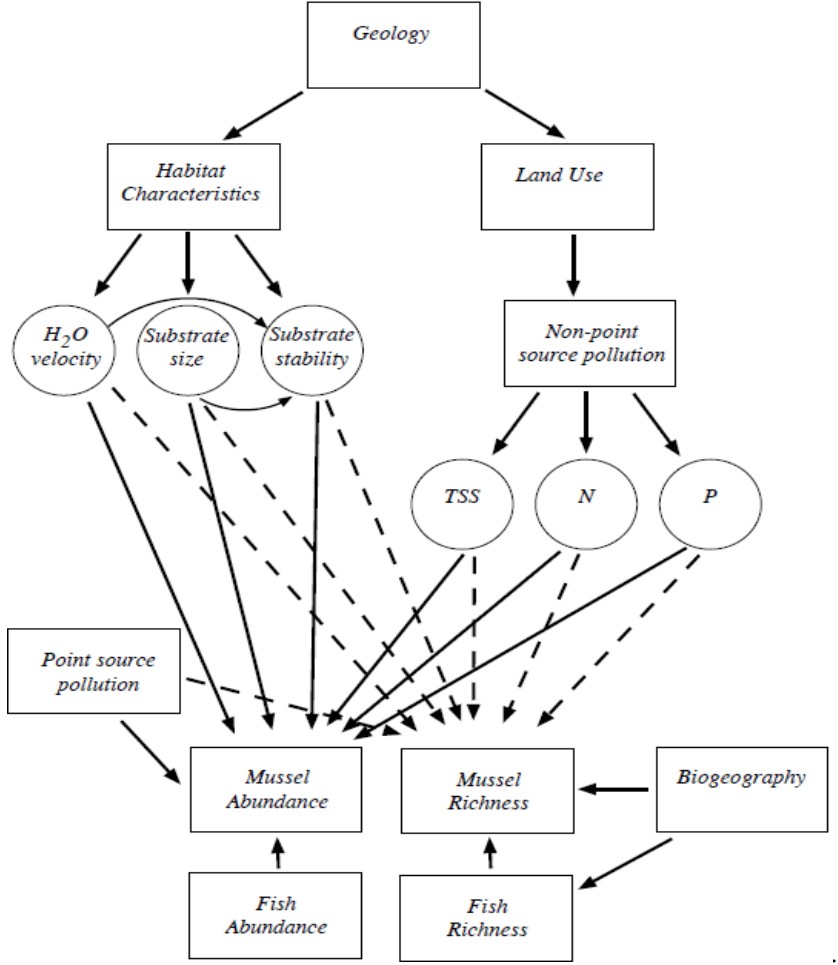

**Figure 2.** Drivers of change for freshwater mussels.

However, the continued imperiled status of freshwater mussels and their documented link with water quality suggest that a conservation strategy focused on clean water policies could be a plausible approach. Designing and incentivizing an effective conservation strategy based on clean water goals requires an understanding of the links between mussels, pollutants, sources of pollution, and regulations, which are discussed below.

*3.3. Mussels, Pollutants, and Water Quality Standards*

As one of the most sensitive aquatic invertebrates, mussels are affected by several parameters affecting water quality, which can be broadly divided into seven categories:

1.  Nutrients: TN, TP, and related parameters such as turbidity (https://www.pca.state.mn.us/sites/default/files/wq-iw3-21.pdf (accessed on 5 July 2023)).
2.  Major ions: chloride and related parameters such as water hardness.
3.  Metals such as copper and zinc.
4.  Sunlight, temperature, and flow regimes.
5.  Contaminated sediments.
6.  Emerging chemicals, including CECs and pesticides.
7.  Sewage and treated wastewater effluent.

The existence of water quality standards for pollutants and their enforcement through NPDES permits based on the Clean Water Act is the main method followed to address and control water pollution, mainly from point sources. Therefore, it is helpful to compare Minnesota's current water quality standards for major toxicants of freshwater mussels with corresponding federal criteria, and mussel tolerance values from the toxicological literature in order to provide a perspective on what is needed in terms of standard development

and regulatory management of point sources to make the state's water quality protective of mussels.

### 3.3.1. Nutrients

Freshwater mussels are more sensitive to nutrient pollution than most other macroinvertebrates [105]. As mussels take up nutrients from the water column, excessive nutrients can affect mussels by impeding early life development. However, a complex indirect pathway exists through algae or phytoplankton. Mussels filter phytoplankton from the water column for food and excrete dissolved nitrogen and ammonia, which the phytoplankton can absorb [37]. However, low filtration rates and/or increased phytoplankton blooms can lead to a decline in mussel numbers due to oxygen deficiency and/or the emergence of toxic cyanobacteria [106]. Increased nutrient loadings from point and non-point sources can lead to such a decline in mussels, in turn leading to a loss in the ecosystem services they provide. Water pollution from excessive nitrogen loads takes the forms of nitrate ($NO_3^-$), nitrite ($NO_2^-$), and ammonia ($NH_3$), while phosphorus exists either in dissolved forms such as orthophosphorus) or particulate (attached to or contained in organic matter and sediment) form (https://cms5.revize.com/revize/columbiaheights/document_center /Stormwater/Phosphorus_201412041423196128.pdf (accessed on 5 July 2023)).

**Ammonia sensitivity.** Freshwater mussels are among the most sensitive aquatic species to ammonia pollution. The USEPA considered freshwater mussels in its 1999 chronic and acute ammonia aquatic life criteria. These criteria are expressed in units of total ammonia nitrogen (TAN) comprising both ammonium ions: ($NH_4^+$) and unionized ammonia: ($NH_3$-N), based on given temperature and pH values (chronic: 4.56 mg TAN/L; acute: 24 mg TAN/L, both at pH 7 and 20 °C) (https://www.epa.gov/sites/production/files/201 5-08/documents/aquatic-life-ambient-water-quality-criteria-for-ammonia-freshwater-20 13.pdf (accessed on 5 July 2023)). Laboratory studies on ammonia tolerance levels for larval and juvenile freshwater mussels found tolerance values in the range 0.37–0.67 mg total ammonia N/L, (i.e., TAN/L) for growth, and 0.37–1.2 mg TAN/L for survival, which, is lower than the 1999 USEPA chronic and acute ammonia criteria [107,108]. A later study investigated the effect of pH on the toxicity of ammonia to juvenile fatmucket mussels (*Lampsilis siliquoidea*), andcompared results with experiments on two other benthic invertebrates, amphipods (*Hyalella azteca*) and oligochaetes (*Lumbriculus variegatus*). The results showed that juvenile mussels were more sensitive to ammonia toxicity than other tested organisms and that mussel sensitivity to ammonia increased with increasing pH [109]. Between the components of total ammonia, unionized ammonia is more toxic to freshwater mussels. It has been documented that unionized ammonia increases markedly with increases in pH, leading to an increase in the toxicity of total ammonia to freshwater mussels. As more nutrients are released into freshwater systems, these waters could become more eutrophic from excessive algal growth, which in turn would lead to an increase in water column pH from photosynthetic activity, resulting in waters less conducive for mussel survival [110].

Based on studies such as these, the USEPA's 2009 Ambient Water Quality Criteria (AWQC) for ammonia, included newly available data on the toxicity of ammonia to freshwater mussels, comprising 67 genera and 12 species compared to 34 genera considered in the 1999 AWQC [111]. The USEPA revised the 2009 criteria in 2013 based on new toxicity data (https://www.federalregister.gov/documents/2013/08/22/2013-20307/final-aquat ic-life-ambient-water-quality-criteria-for-ammonia-freshwater-2013 (accessed on 5 July 2023)), and a reanalysis of data considered in the 1999 and 2009 ammonia criteria documents. The resulting ammonia criteria, which accounted for data on unionid mussels [112], are: chronic: 1.9 mg TAN/L and acute: 17 mg TAN/L, both at pH 7 and 20 °C [113]. Minnesota currently has a chronic ammonia standard expressed in μg/L of $NH_3$-N that is based on the USEPA 1980 criteria. The current criteria for ammonia at both the federal (USEPA-2013 guidance) and state level (MN-1981 water quality standards) are provided in Table 1.

**Table 1.** USEPA and Minnesota ammonia criteria.

| | Acute | Chronic |
|---|---|---|
| USEPA (pH = 7, temp = 20 °C) | 17 mg TAN/L | 1.9 mg TAN/L |
| Minnesota | n/a | 16 µg/L NH$_3$-N (Class 2A) and 40 µg/L NH$_3$-N (Class 2B) |

The USEPA chronic ammonia criteria at different pH and temperature values are provided in Table 2. The Minnesota chronic ammonia criteria for Class 2A (cold water aquatic communities) and Class 2B (cool or warm water aquatic communities) waters in terms of TAN/L are provided in Tables 3 and 4, respectively, for convenient comparison. Comparing the Minnesota chronic standard for ammonia in Tables 3 and 4 to the USEPA ammonia criteria in Table 2, we find that the Minnesota criteria for Class 2A waters (Table 3) are protective of mussels at certain temperature–pH combinations, i.e., the MN cold water criteria are protective of mussels at temperatures higher than 100 °C and pH $\geq$ 7.5 (Table 3), while the criteria for Class 2B waters (Table 4) are not fully protective of mussels at any temperature–pH combination shown.

**Table 2.** USEPA 2013 chronic ammonia criteria as mg TAN/L with mussels present.

| pH/Temp | 0 | 5 | 10 | 15 | 20 | 25 | 30 |
|---|---|---|---|---|---|---|---|
| 6.5 | 4.921564 | 4.921564 | 4.056048 | 2.938347 | 2.128644 | 1.542067 | 1.117128 |
| 7.5 | 3.221817 | 3.221817 | 2.655222 | 1.923539 | 1.393481 | 1.009487 | 0.731309 |
| 8.5 | 0.804085 | 0.804085 | 0.662677 | 0.480067 | 0.347778 | 0.251943 | 0.182516 |

**Table 3.** MN cold water chronic total ammonia nitrogen standards as mg TAN/L (converted from 0.016 mg/L unionized ammonia).

| pH/Temp | 0 | 5 | 10 | 15 | 20 | 25 | 30 |
|---|---|---|---|---|---|---|---|
| 6.5 | 61.4293248 | 40.62434 | 27.2626 | 18.55231 | 12.79335 | 8.934211 | 6.314915 |
| 7.5 | 6.15733248 | 4.076834 | 2.74066 | 1.869631 | 1.293735 | 0.907821 | 0.645892 |
| 8.5 | 0.63013325 | 0.422083 | 0.288466 | 0.201363 | 0.143774 | 0.105182 | 0.078989 |

**Table 4.** MN cool and warm Water chronic total ammonia nitrogen standards as mg TAN/L (converted from 0.04 mg/L unionized ammonia).

| pH/Temp | 0 | 5 | 10 | 15 | 20 | 25 | 30 |
|---|---|---|---|---|---|---|---|
| 6.5 | 153.573312 | 101.5609 | 68.15649 | 46.38078 | 31.98338 | 22.33553 | 15.78729 |
| 7.5 | 15.3933312 | 10.19209 | 6.851649 | 4.674078 | 3.234338 | 2.269553 | 1.614729 |
| 8.5 | 1.57533312 | 1.055209 | 0.721165 | 0.503408 | 0.359434 | 0.262955 | 0.197473 |

Minnesota currently does not have an acute standard for ammonia. Acute standards are based on pollutant concentrations that lead to mortality over short durations, while chronic standards are based on pollutant concentrations that would affect growth, survival, and reproduction over the longer term. Therefore, having both types of standards can help more comprehensively manage the multiple goals of clean water provisioning, including protection of beneficial uses, conservation of imperiled species such as freshwater mussels, and regulatory management of permitted discharges.

Consequently, the Minnesota Pollution Control Agency (MPCA) is revising its ammonia standard to adopt the USEPA 2013 ammonia criteria. The standard values would change according to temperature and pH conditions to reflect the underlying relationship

between these variables and ammonia toxicity, as reflected in Figure 3, which translates the USEPA 2013 criteria to Class 2 waters in Minnesota. The resulting standard accounts for ammonia sensitivity to a wider variety of aquatic life, including freshwater mussels in Minnesota lakes and streams [114].

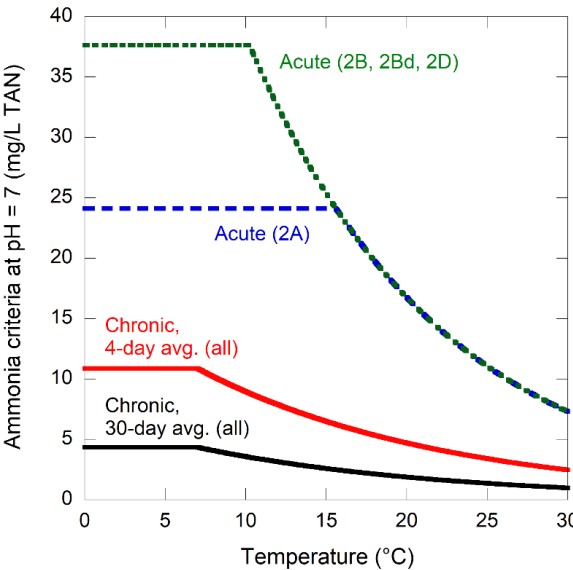

**Figure 3.** Recommended ambient water quality criteria for the protection of aquatic life (USEPA, 2013 [113]) and their translation to Class 2 waters in Minnesota. Note: Numeric values are extrapolated across a temperature gradient at pH = 7. Source: Adapted from MPCA Draft Ammonia TSD, 2022.

The USEPA 2013 ammonia criteria are expected to be protective of mussels in Minnesota based on a review of available ecotoxicological data. However, because many species of mussels are critically endangered, it helps to be mindful of the threats to their survival based on emerging drivers, such as climate change and increased human activity, that could affect the interaction between pollutants and water quality parameters.

For example, excessive unionized $NH_3$ can prevent juvenile mussel recruitment, with a significant effect on the total mussel population [13]. This effect is worsened if increased nutrient loadings from human activity led to high rates of photosynthesis, which could increase pH, in turn leading to larger amounts of toxic unionized $NH_3$, which is particularly harmful for juvenile mussels. Apart from pH, temperature can also affect mussels' sensitivity to ammonia. As noted in the USEPA's 2013 ammonia criteria document, "In contrast to the pH–toxicity relationship, which applies to both vertebrates and invertebrates, the temperature–ammonia toxicity relationship only applies to invertebrates. Based on the results of the 1999 reanalysis of this relationship, it was determined that ammonia toxicity for invertebrates decreases with decreasing temperature to a temperature of approximately 7 °C, below which the relationship ends [115]. As temperature increases in the aquatic environment, the toxicity of ammonia to mussels increases. The effect of temperature combined with pH on ammonia sensitivity was reported in a study of twenty river basins in China, which showed higher ammonia toxicity in mussels at sites with good water quality in summer and autumn, when water temperature and pH are higher [116].

Recent research has noted higher ammonia sensitivity of freshwater mussels under certain conditions. For example, the $LC_{50}$ value or Lethal Concentration 50 (the concentration of the chemical at which 50% of test animals are reported killed during the observation period) for acute ammonia toxicity for the critically endangered freshwater mussel *Pseudunio auricularius* in the disturbed and polluted Ebro basin of the Iberian Peninsula was found to be 7.53 mg of TAN/L based on 96 h acute toxicity tests [117]. Chronic toxicity estimates

for two species of freshwater mussels (the lotic *Velesunio* sp. and the lentic *Velesunio angasi*) in tropical northern Australia, characterized by soft waters (water hardness of <5 mg/L), were found to be 7–9.2 mg of TAN/L for V. angasi and 11.3 mg of TAN/L for Velesunio sp. at pH 6.0 and 27 ± 0.5 °C [118]. When normalized to average temperate conditions of pH 7 and 20 °C, these freshwater mussel species were still found to be more sensitive than 8 of 16 other temperate, and 7 of 9 other tropical fish and invertebrate species. These differences in chronic ammonia toxicity were determined to be mainly due to the lower ionic strength of tropical fresh waters, based on the results of a study involving toxicity tests on six tropical species in Australia [119]. These effects of ammonia toxicity can significantly affect freshwater mussels, altering the valuable services provided by mussels to freshwater ecosystems, and as such should be considered as part of an effective mussel conservation strategy.

**Nitrate and Phosphorus sensitivity.** A non-monotonic function was proposed to describe the multiple and complex pathways linking nutrients and mussels, by [13] (Figure 2 from [13] reproduced here as Figure 4). In thisrelationship, mussel abundance increases as total nitrogen and total phosphorus loads (TN and TP) increase from low levels and where fish hosts are more abundant, then reach a peak, after which proliferation of algal blooms cause eutrophication and may cause production of unionized ammonia ($NH_3$), which leads to a decline in juvenile mussels, and then in adult mussels. The parameters of this generic curve vary across mussel species.

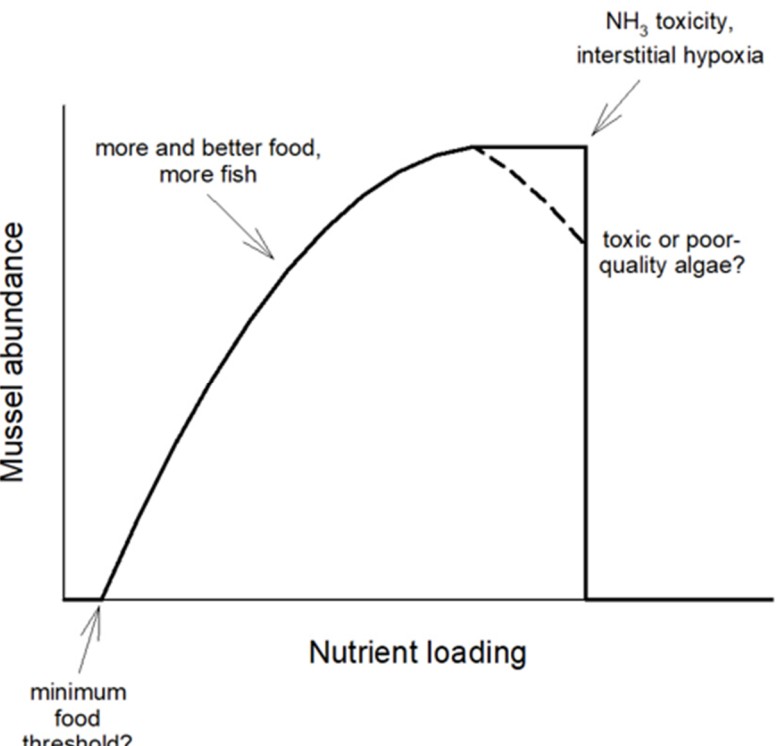

**Figure 4.** Freshwater mussel abundance, phytoplankton, and nutrient loading. Source: Strayer et al., 2014 [13].

Nitrate pollution is one of the main types of nutrient pollution nationally and globally and is recognized as being harmful to aquatic organisms. Recently, Minnesota revised a draft technical support document that developed endpoints for nitrate toxicity in a variety of aquatic organisms, including freshwater mussels [120].

Nitrate, in addition to its indirect effect through algae, can have a toxic effect on freshwater mussels on its own. For example, experimental evidence of nitrate toxicity to mussels from the catchment of the Lužnice River, Czech Republic, was provided by [121]. The results of a logistic regression based on mussel occurrence and nitrate-nitrogen concentration

data showed that the spatial co-occurrence of five native mussel species declined with increasing nitrate concentrations, in agreement with earlier studies in Central European streams. Laboratory tests for acute (96 h) nitrate toxicity on adult freshwater mussels have found $LC_{50}$ values ranging between 357–937 mg of $NO_3$-N/L for the species *L. siliquoidea* and *L. nervosa* [122]. Nitrate toxicity in the early life stages of freshwater mussels, particularly the glochidia (larval stage) and juvenile forms, was reported by [92]. In this study, glochidia of two freshwater mussel species, *L siliquoidea* and *Lampsilis fasciola*, were exposed to nitrate solutions for 24 h (acute exposure) before being transported to a host fish. The results of this test showed a 28–35% reduction in juvenile production in *L. siliquoidea* from exposure to 11 and 56 mg of $NO_3$-N/L, while no change was seen in *L. fasciola*, showing differences between these closely related species. More recent acute toxicity tests with glochidia of four freshwater mussel species (*Hamiota altilis*, *L. fasciola*, *L. siliquoidea*, *Utterbackiana suborbiculata*) have shown median effective concentrations ($EC_{50}$s) at 524–904 mg/L of $NO_3$-N/L in moderately hard water, i.e., 80–100 mg/L of $CaCO_3$ [123] and 665 mg/L for *L. siliquoidea* on its own [124]. Chronic 28-day toxicity tests for nitrate for *L. siliquoidea* found 20% effective concentrations ($EC_{20}$s) at 17 mg of $NO_3$-N/L [124].

Phosphorus is typically the most limiting nutrient for algal growth in lakes and has a strong correlation with chlorophyll-*a* [125]. These indirect mechanisms affecting aquatic life can be significant in freshwater systems. Phosphorus negatively impacts freshwater mussels in three ways. Excess phosphorus loading leads to increased phytoplankton growth, which in turn leads to reduced DO levels [126], which is detrimental to mussel survival. A study on the oxygen consumption (OC) rates of nine freshwater mussel species found that levels of dissolved oxygen (DO) concentrations in streams needed to be >2–4 mg/L for six of the species tested (*Pyganodon grandis*, *Amblema plicata*, *Quadrula pustulosa*, *Elliptio complanata*, *Elliptio fisheriana*, and *Elliptio lanceolata*), 3.5–4 mg/L and >4 mg/L, respectively, for two more sensitive species (*Pleurobema cordatum* and *Villosa constricta*), and >6 mg/L for the most sensitive species (*Villosa iris*) in order to maintain normal OC under a temperature of 24 degrees Celsius [127]. In addition, excess phosphorus can trigger toxic cyanobacterial blooms, specifically when N is low [128], which reduces food quality for mussels. Finally, phosphorus leads to the production of toxins such as microcystins [129] as well as unionized $NH_3$ from cyanobacteria, which is highly toxic to juvenile mussels [113].

A water quality study of five rivers using principal component analysis of sampling events for 1998 and 2004 showed that the abundance of *L. fasciola* was negatively correlated with the concentration of TP, nitrate, and nitrite ($NO_2^-$), total Kjeldahl nitrogen (TKN), and turbidity [93]. Mussels were found to be most abundant at TP < 0.05 mg/L and nitrate and nitrite < 3 mg/L, while no live mussels were found at TP > 0.10 mg/L and turbidity levels > 8 Jackson Turbidity Units (JTU) (Turbidity: https://en.wikipedia.org/wiki/Turbidity (accessed on 5 July 2023)). The authors concluded that as higher N and P loadings lead to higher turbidity, it is plausible that the impacts of these contaminants on *L. fasciola* are linked.

Minnesota currently does not have stand-alone aquatic life standards for nitrate or phosphorus. However, as noted previously, Minnesota recently revised a draft technical support document (TSD) for nitrate which developed toxicity endpoints for a variety of aquatic organisms, including freshwater mussels [120]. Canada's 2012 aquatic life guidelines provide criteria for both chronic and acute toxicity values for nitrates, the latter based on freshwater mussel exposure data (Table 5). The chronic value (3 mg/L) (https://www2.gov.bc.ca/assets/gov/environment/air-land-water/water/waterquality/water-quality-guidelines/approved-wqgs/bc_env_nitrate_waterqualityguideline_overview.pdf (accessed on 5 July 2023), https://www.ccme.ca/fr/res/2012-nitrate-cwqg-scd-1470-en.pdf (accessed on 5 July 2023)) is low enough to be considered protective according to ongoing research on nitrate toxicity to mussels [93,124].

**Table 5.** Canadian water quality guideline on nitrate ions for protection of aquatic life.

|  | NO$_3$-mg/L Chronic | NO$_3$-mg/L Acute |
|---|---|---|
| Freshwater | 13 | 550 |
|  | 3 | 124 |

Minnesota's Lake and River Eutrophication Standards, LES and RES, respectively, [130,131], are based on excess phosphorus as the causal agent of eutrophication associated with conditions of excess chlorophyll-*a*, DO flux, and biochemical oxygen demand (BOD), all of which are harmful to aquatic life and reduce recreation quality. The RES are more explicitly based on aquatic life compared to the LES [See https://www.pca.state.mn.us/sites/default/files/wq-s6-08.pdf (accessed on 5 July 2023)]. These criteria are not based directly on freshwater mussels, and incorporate data on many kinds of aquatic life, including fish and macroinvertebrates. The LES and RES include a cause criterion (i.e., TP), and response criteria comprising stressors (i.e., Chl-*a*, BOD, and daily DO flux) [130,131]. The RES criteria, which are more explicitly based on aquatic life, are shown in Table 6.

**Table 6.** River eutrophication criteria by river nutrient region in Minnesota.

|  | Nutrient | Stressors | | |
|---|---|---|---|---|
| **Region** | **TP (μg/L)** | **Chl-*a* (μg/L)** | **DO Flux (mg/L)** | **BOD (mg/L)** |
| North | ≤50 | ≤7 | ≤3.0 | ≤1.5 |
| Central | ≤100 | ≤18 | ≤3.5 | ≤2.0 |
| South | ≤150 | ≤40 | ≤5 | ≤3.5 |

Turbidity is addressed by the criteria for total suspended sediments (TSS) in part 3 of the SONAR for the RES (MPCA document #: wq-rule4-06e, Eutrophication and TSS SONAR, Book 3: https://www.pca.state.mn.us/sites/default/files/wq-rule4-06g.pdf; (accessed on 5 July 2023)). Streams that exceed both the cause and response criteria in Table 6 are likely to be considered "impaired" for RES and are regulated through NPDES permits with total maximum daily loads (TMDLs) [131]. They are additionally regulated through reasonable potential (RP) analyses to ensure that beneficial uses are met; such analyses include effluent limits in permits where there is RP to exceed a WQS [131].

The Minnesota 2022 Impaired Waters list [97] found that 40% of Minnesota's lakes and streams are impaired by conventional pollutants [132], including parameters for eutrophication. Conventional pollutants refer to a list of pollutants commonly used to assess the state's waterbodies by watershed based on a 10-year assessment cycle, called Intensive Watershed Monitoring (https://www.pca.state.mn.us/air-water-land-climate/watershed -approach-to-water-quality (accessed on 5 July 2023)). While controlling eutrophication is beneficial to mussels, which are particularly sensitive to ammonia from toxic algal blooms, the levels of stressors, including TP, that are deemed safe in RES were not developed incorporating data on freshwater mussels [131].

The 2004 Canadian guidance framework document for phosphorus states: "The first response of an aquatic system to phosphorus additions is increased plant and algal productivity and biomass. Although this may be desirable in some cases, beyond a certain point, further phosphorus additions to phosphorus-limited systems can cause undesirable effects ... When the excessive plant growth includes certain species of cyanobacteria, toxins may be produced, causing increased risk to aquatic life, livestock, and human health" ([133], page 2). The framework document provides TP trigger ranges for Canadian lakes and rivers, which are shown in Table 7.

**Table 7.** Total phosphorus trigger ranges for Canadian lakes and rivers.

| Trophic Status | Canadian Trigger Ranges TP (µg/L) |
|---|---|
| Ultra-oligotrophic | <4 |
| Oligotrophic | 4–10 |
| Mesotrophic | 10–20 |
| Meso-eutrophic | 20–35 |
| Eutrophic | 35–100 |
| Hyper-eutrophic | >100 |

In the absence of a stand-alone aquatic life standard for TP, and further considering that the RES was not explicitly developed to protect freshwater mussels, it is helpful to recall the results for freshwater mussel abundance found in the study of five rivers discussed before. As noted, mussels were found to be most abundant at TP < 0.05 mg/L and both nitrate and nitrite < 3 mg/L (in agreement with the Canadian chronic guideline for nitrate), while no mussels were found alive at TP > 0.10 mg/L and turbidity levels > 8 Jackson Turbidity Units (JTU) (Turbidity: https://en.wikipedia.org/wiki/Turbidity (accessed on 5 July 2023)) [93]. Based on freshwater mussel survey data from the MDNR (Figure 1), southern Minnesota has generally higher mussel extirpation rates as well as a larger contribution to pollution from nutrients; therefore, the corresponding TP standard of 0.15 mg/L for the South River nutrient region (Table 6) may not be sufficiently protective.

It follows that for nutrients generally, current water quality standards in Minnesota may not be sufficiently protective of mussels, leaving scope for Minnesota's proposed adoption of the 2013 USEPA ammonia criteria and revised nitrate TSD to be a step in the positive direction for freshwater mussel conservation [114,120].

3.3.2. Major Ions

Excessive loading of major ions or 'salts' from human activity in surface waters comprises cations such as sodium, potassium, calcium, and magnesium (Na, K, Ca, Mg) and anions such as chlorides, sulfates, and carbonates (Cl, $SO_4$, $HCO_3$). These salts contribute to salinization of surface water globally [134]. Consequently, they are a serious concern for sensitive aquatic life such as freshwater mussels in the northeastern U.S. [135,136], and in Great Lakes states such as Minnesota [137]. In addition, they can adversely affect drinking water supplies [138]. The main anthropogenic sources for cations are mountaintop coal mining, valley filling, and mine drainage, while anions may result from road salt, alkali production, mine drainage, and water softeners, the latter further contributing to increased toxicity of salts to aquatic life by reducing water hardness [139].

Major ions in various combinations of cations and anions contribute to total dissolved solids (TDS), which increases the specific conductivity of water, another parameter of concern for aquatic life [140]. In chronic laboratory tests conducted with four invertebrate species (a cladoceran (*Ceriodaphnia dubia*), a freshwater mussel (*L. siliquoidea*), an amphipod (*H. azteca*), and a mayfly (*Centroptilum triangulifer*)) in water simulating Appalachian streams contaminated with either alkaline mine drainage from mountaintop mining with elevated Mg, Ca, K, $SO_4$, and $HCO_3$ ions or neutralized mine drainage with elevated Na, K, $SO_4$, and $HCO_3$ ions, both were toxic to freshwater mussels at the lowest tested concentration of 10%. The former was toxic to all except the cladoceran, and the latter was toxic to all except the mayfly, showing that elevated levels of total dissolved solids (TDS) based on the ionic composition, are particularly toxic to freshwater mussels, while being toxic to several other invertebrates [141].

The acute toxicity of sodium chloride to glochidia of four species of freshwater mussels based on the 24-h $EC_{50}$ values, was found to be in the range 1265–1559 mg Cl/L for natural water (hardness 278–322 mg $CaCO_3$/L) and much lower for water reconstituted with salt (hardness: 100 mg $CaCO_3$/L): 113–1430 mg Cl/L (285 mg Cl/L for glochidia of *L.*

*fasciola*) [89]. These findings show that glochidia of freshwater mussels are very sensitive to chloride toxicity, and additionally, are significantly more sensitive when hardness is lower [89]. In addition, chloride levels in streams with endangered mussel habitats in the Great Lakes basin in Canada can exceed 1300 mg Cl/L, which is toxic to glochidia regardless of water hardness, raising a similar concern for freshwater mussel habitats in the Great Lakes region of the U.S. affected with salt pollution [89,137]. In another study testing the acute (96 h) toxicity of five species of juvenile mussels to ten chemicals, including chloride, the mussels had similar sensitivity based on $EC_{50}$s for most of the tested chemicals. In addition, the sensitivity of *L. siliquoidea*, a commonly tested mussel, compared well with those of the other species, indicating *L. siliquoidea* to be a representative species for toxicity testing [90]. Chronic toxicity tests for sodium chloride or potassium chloride (NaCl or KCl) on juveniles of *L. siliquoidea* found EC20s at 264 mg/L for NaCl and 8 mg/L for KCl, showing these mussels to be the most sensitive species to KCl toxicity and the second-most sensitive to NaCl toxicity among all freshwater organisms [142].

Mussels have been documented to be even more sensitive to a combination of salts [143]. As most chloride loading from WWTPs is due to home water softener use, the corresponding effluent can contain a high amount of chloride, which is toxic to aquatic life [17]. Current research has documented that the type of salt matters for the extent of toxicity to freshwater aquatic life, with sodium chloride being more toxic compared to calcium or magnesium chloride [144]. Chloride impairment of waterbodies is a growing problem in Minnesota, with 50 waters currently impaired and 75 at risk of impairment [145].

Minnesota has adopted the USEPA's national ambient chloride criteria, with an aquatic life standard of 230 mg/L [146] and a secondary drinking water standard of 250 mg/L (https://www.epa.gov/sdwa/drinking-water-regulations-and-contaminants (accessed on 5 July 2023)). Canada has aquatic life guidelines for chloride that include chronic and acute toxicity data for mussels [147] (http://ceqg-rcqe.ccme.ca/download/en/337 (accessed on 5 July 2023)), as shown in Table 8.

**Table 8.** Canadian water quality guideline for chlorides ion for protection of aquatic life.

|  | Cl-mg/L Chronic | Cl-mg/L Acute |
| --- | --- | --- |
| Freshwater | 120 | 640 |

However, current research has found that these North American chloride standards may not be protective of aquatic life due to the adverse effect of freshwater salinization from chloride pollution on zooplankton populations, which can in turn disrupt ecosystem functions related to lake food webs [136,148].

For sulfate, acute and chronic values for *Lampsilis abrupta* have been recorded at 2362 and 1795 mg $SO_4$/L [149]. Recent laboratory tests documented acute and chronic toxicity of sulfates in the pinkmucket mussel *L. abrupta* and three other aquatic organisms: a cladoceran, a midge, and a fish (the fathead minnow) [149]. The mussel ($LC_{50}$: 2362 mg $SO_4$/L) and the cladoceran ($LC_{50}$: 2441 mg $SO_4$/L) were found to be more acutely sensitive to sulfate, while the fish was the most chronically sensitive in the embryonic stage (305–477 mg $SO_4$/L for survival), followed by the mussel ($LC_{50}$: 1759 mg $SO_4$/L). The same study found that increased potassium (from 1 to 3 mg K/L) significantly reduced the toxicity of sulfate to fish embryos. Overall, these results highlight the toxicity of sulfate to aquatic life, including mussels, and the importance of considering complete toxicity data as well as the mitigating effect of potassium on sulfate toxicity when developing guidance values for sulfate.

### 3.3.3. Metals

Deposition of heavy metals such as cadmium, mercury, and lead, mainly from anthropogenic sources, has historically led to increases in metal concentration in waterbodies significantly in excess of natural background levels. This is particularly evident after the

industrial revolution, as indicated by analysis of sediments in ice cores, lakes, and peat bogs in North America [150]. While atmospheric deposition has declined in some areas of the northern hemisphere in the 21st century thanks to improvements in pollution control, metal concentrations can be elevated in many waterbodies [151,152] due to legacy concentrations in biota, potentially impacting biotic processes and interacting with exogenous drivers such as climate change [153]. These elevated concentrations are toxic to mussels, which are exposed to metals dissolved in the water column or deposited in sediment and may bioaccumulate these toxins over their life cycle, resulting in a variety of effects including death and/or changes in growth, functionality, and behavior [154]. In addition, metal deposition, particularly mercury deposition, can indirectly affect mussels by affecting their host fish species [153,155].

A review of metal toxicity to mussels has found that a possible cause of the widespread decline in freshwater mussels in North America could be chronic exposure to heavy metals such as cadmium, copper, lead, mercury, and zinc [66]. Freshwater mussels have been foundto be particularly sensitive to mercury toxicity, with acute toxicity values for mercury ranked higher than those for cadmium and zinc [156]. In laboratory tests with the rainbow mussel (*V. iris*) mercury was confirmed to be toxic to both glochidia and juvenile mussels, with acute toxicity values in the range of 14–107 µg Hg/L for glochidia, and 99–162 µg Hg/L for juvenile mussels. In the chronic test, juvenile mussels stopped growing at 8 µg Hg/L. While these recorded toxicity values of mercury are much higher than Minnesota's mercury WQS, there are multiple pathways by which mercury is capable of entering waterbodies, and it is currently the leading cause of surface water impairments in Minnesota [97].

Acute and chronic toxicity of lead, cadmium, and zinc in the early life stages of freshwater mussels (glochidia and juveniles) in laboratory studies, was confirmed by [157], while the same for acute and chronic toxicity of copper (2 and 12 µg of Cu/L, respectively), was confirmed by [91,108]. These results show that mussels are highly sensitive to all of the above metals compared to other freshwater species, except for chronic exposure to cadmium and zinc, for which the comparison is moderate. Current USEPA aquatic life criteria are protective of mussels against cadmium and lead toxicity, but not against zinc toxicity, for which chronic values were recorded in the range (63–68 µg Zn/L) [157]. In addition, current USEPA criteria for copper are not protective of freshwater mussels [91] (https://www.epa.gov/wqc/national-recommended-water-quality-criteria-aquatic-life-criteria-table (accessed on 5 July 2023)). It is noteworthy that the toxicity of certain metals may depend on other factors, such as pH, ambient ions (CA, Mg/water hardness), and dissolved organic carbon (DOC) concentrations [157–160].

Minnesota has acute and chronic standards for cadmium (7.8 and 1.1 µg/L), copper (35 and 9.8 µg/L), lead (164 and 3.2 µg/L), and zinc (234 and 106 µg/L). It also has a human health-based chronic standard for mercury (6.9 ng/L) at a water hardness of 100 (https://www.revisor.mn.gov/rules/7050.0222/ (accessed on 5 July 2023)). However, none of these standards are based on mussel toxicity information.

3.3.4. Sunlight, Temperature, and Flow Regime Sensitivity

Evidence of the strong links between nutrient loads and mussel activity, was provided in a study conducted in the St. Croix River system [161]. The study results connected nutrient loads with mussel growth and survival through phytoplankton abundance and other key drivers such as sunlight, temperature, and flow/turbulence regimes. Increased nutrient loadings of TN and TP led to increases in cyanobacteria, which in turn increased the particle size of food available to mussels, thereby altering its quality. Factoring in the effects of light, temperature, and flow regimes, the study found that juvenile mussel recruitment is likely to decline in a nutrient-rich environment, particularly in slow-moving streams and confined waterbodies such as lakes and reservoirs.

As mussels initially depend on fish hosts to complete their life cycle, they are affected by both their own tolerance to environmental stressors and by the tolerances of their hosts [50,162]. This general trend is reversed for the freshwater mussel *Anodonta*

*anatina* and the European Bitterling, where it has been documented that high concentrations of P (>500 μg/L) are detrimental to the sustenance of bitterling larvae by their mussel hosts [163]. Variations in mussel thermal tolerance do not depend on any specific factor, most likely due to complex interactions among multiple factors, which highlights the importance of examining the impacts of global changes on aquatic organisms at the community level, as reported by [164]. The authors conducted a meta-analysis on the thermal sensitivity of eight species of freshwater mussels and their host fishes to determine whether these aquatic communities are at risk from climate change. In 62% of mussel–host fish comparisons, mussels were more thermally tolerant than their hosts (3.4 °C mean difference; range = 0.2–6.8 °C), suggesting that certain mussels may be more stenothermic than tolerance criteria indicate. Being more stenothermic would increase mussels' vulnerability to thermal stress due to climate change, which could impede life cycles and survival due to a lack of available hosts. Thermal tolerances for both glochidia and juvenile forms of eight freshwater mussel species were tested and found to vary for glochidia but not for juveniles, while not being affected by acclimation to higher temperatures in either life stage [165]. These results show that certain mussel species may already be living close to their upper thermal tolerances, increasing their vulnerability to climate change.

As mussels, particularly juvenile mussels, live submerged in sediment for long time periods, a study tested the thermal tolerance in sediment of juvenile mussels from four mussel species, for a range of summer temperatures and compared the results with standard thermal sensitivity tests without sediment (water-only tests) [166]. The study found that the presence of sediment alone did not affect mussel thermal sensitivity for juveniles, showing that streambeds may not always be protective thermal refuges for mussels. The effect of climate change on mussels was examined by exposing four mussel species (*Amblema plicata*, *Elliptio complanata*, *Fusconaia flava,* and *Lampsilis cardium*) to a range of elevated temperatures (20, 25, 30, 35 °C) in 21-day laboratory tests [167]. The results of this study showed that elevated water temperatures worsened physiological responses in mussels in terms of a general increase in oxygen consumption and ammonia excretion rates, significant variations in the oxygen/nitrogen ratio (O:N), and variation in tissue condition based on temperature and species. The authors concluded that higher temperatures could affect metabolic rates in mussels, which in turn could affect key biological processes such as survival, growth, and reproduction.

### 3.3.5. Sediment Toxicity

Contaminants in sediments can be toxic to freshwater mussels. These contaminants, including toxic compounds in sediments, turbidity in suspended sediments, and nutrients in sediments, can affect mussels in multiple ways. Sediments can carry contaminants that can be toxic to freshwater mussels into waterbodies. Sediments containing agricultural runoff can lead to excess loading of nutrients and ammonia in streams, while activities such as mining can lead to sediments contaminated with heavy metals and major ions.

Chronic toxicity tests using sediments from the basins of the upper Tennessee and Cumberland rivers (sites impacted by coal and gas mining) on two freshwater mussel species (the rainbow mussel, *V. iris*, and wavy-rayed lampmussel, *L. fasciola*) along with an amphipod (*H. azteca*) and a midge (*Chironomus dilutes*) showed that these sediments significantly affected the survival, biomass, and size of one or more species, particularly the mussels (50-63% of samples) [168]. Heavy metals, polycyclic hydrocarbons (PAHs), and major ions in sediments were the cause of this toxicity. The authors found that toxicity to mussels increased with increases in probable effective concentration (PEC) quotients for the metals and total PAHs, even though the concentration of metals and PAHs in the sediments were below PEC levels identified as safe for aquatic life. This result implies that the current sediment toxicity guideline of 1 based on PECs is not protective of freshwater mussels. More recent chronic laboratory tests on juvenile mussels using sediment samples from the Clinch River in Virginia and Tennessee showed that these riverine sediments significantly affected mussel survival and biomass [169]. The authors found sediments to

be contaminated with metals such as manganese, polycyclic hydrocarbons, and ammonia. Among these contaminants, manganese was found to be detrimental to both the survival and biomass of mussels, while ammonia was found to be detrimental to survival and total organic carbon to biomass. The authors associated the elevated levels of manganese and ammonia in the river sediments to current land uses in the watershed, which include fossil fuel mining and agriculture.

A process-based interaction model to highlight the complex links between freshwater mussels and environmental stressors such as suspended sediments and phytoplankton in rivers and streams, was developed by [15]. This model's underlying assumption is that increased streamflow from agricultural watersheds causes higher sediment loads and phytoplankton, which collectively affect mussel population density through food limitation. The model was applied to the basins of the Minnesota and St. Croix Rivers and used to predict changes in mussel abundance, including a scenario-based sensitivity analysis where mussel population density showed critical threshold responses to long-term changes in suspended sediment concentration. This finding suggests that regulating sediment and nutrient loads in streams would be beneficial for long-term mussel survival, similar to the documented link between nutrient loads and algal growth in the St. Croix River [170]. Criteria for total suspended sediments (TSS) addressing turbidity, are provided in part 3 of the SONAR for the Minnesota RES (MPCA document #: wq-rule4-06e, Eutrophication and TSS SONAR, Book 3: https://www.pca.state.mn.us/sites/default/files/wq-rule4-06g.pdf (accessed on 5 July 2023)).

### 3.3.6. Contaminants of Emerging Concern (CECs)

Contaminants of emerging concern, or CECs, denote a wide variety of chemicals that are increasingly being detected in waterbodies as well as in soils and sediments [171–173] and bound to plastics [174,175]. These pollutants, including "nanoparticles, pharmaceuticals and personal care products (PPCPs), estrogen-like compounds, flame retardants, detergents, and some industrial chemicals" (https://www.epa.gov/columbiariver/chemicals-emerging-concern-columbia-river (accessed on 5 July 2023)), are mainly from anthropogenic sources, including agricultural runoff, air pollution, and wastewater effluent, with wastewater treatment plant discharges being the major sources for PPCPs [176,177]. Microplastics (MPs) are plastic particles up to five millimeters in diameter that include fragments from larger previously broken-down plastic items, such as clothing fibers (acrylic and polyester) and small particles referred to as microbeads (https://en.wikipedia.org/wiki/Microbead (accessed on 5 July 2023))), while nanoplastics (NPs) consist of plastic particles less than 100 nm in diameter. Both can be viewed as CECs, as they are composed of organic polymers that may include chemicals classified as CECs or use CECs as additives to improve properties such as tensile strength, color, and use as flame-retardants [178]. The prevalence of CECs in the U.S. [179] and Minnesota waters (https://www.pca.state.mn.us/water/pollutants-emerging-concern (accessed on 5 July 2023)) and in the Great Lakes system [180,181] represents a concern due to their potentially significant impacts on aquatic life and human health [182]. These chemicals are often endocrine-active and can bioaccumulate in organisms even at very low concentrations, impacting growth, development, and reproductive functions over time [183–185].

According to a recent report by the NOAA's Mussel Watch Program (https://www.fisheries.noaa.gov/inport/item/39400 (accessed on 5 July 2023)), which measured 237 CECs (PPCPs, pesticides and phenols) in the Great Lakes system, 99 (42%) CECs were detected in the tissues of freshwater mussels used as biomarkers for chemical contamination. While CECs were detected at all sites tested, they were more frequently found in sites located in river/harbor areas with known pollution sources including agrochemical runoff and wastewater treatment outfalls. In addition, CECs were found in several 'reference' sites located away from pollution sources, suggesting both the easy transport of these chemicals and their persistence over time [186].

Freshwater mussels are among the most sensitive invertebrate species, are known to bioaccumulate, and feature differential exposure and impacts to pollutants across life stages [187]. High CEC concentrations have been documented in mussels. For example, 145 PPCPs were found in mussels sampled from the Grand River in Ontario, Canada, which receives wastewater effluent from a major WWTP (https://www.water-technology.net/projects/kitchener-wastewater-treatment-plant-upgrade-ontario/ (accessed on 5 July 2023)) [188]. Moreover, freshwater mussels have been recognized to experience severe developmental impacts from CECs, including spawning in male mussels [189]. Laboratory tests confirmed the acute and chronic toxicity of pesticide formulations including technical grade pesticides such as chlorpyrifos and atrazine (21-day exposure median 4.3 mg/l with visible effects at => 3.8 mg/l), as well as acute toxicity of the common pesticide glyphosate to both glochidia and juveniles of *L. siliquoidea* at a concentration lower than any tested aquatic organism (48-h exposure at median 0.5 mg/L) [86–88]. These results demonstrate that freshwater mussels are among the most sensitive aquatic organisms in this context. Chronic toxicity tests with juveniles of *L. siliquoidea* exposed to high concentrations of PPCPs showed a decrease in functions such as feeding and filtering [190]. Freshwater mussels have been documented to bioaccumulate and show toxicity to antimicrobial chemicals and microplastics, with significant declines in functions such as feeding, filtration rates, and enzyme activity [191], ultimately leading to oxidative damage and neurotoxicity either alone or in combination with metals such as mercury [192].

Considering the global spread of CECs in freshwater environments, including MPs and NPs [172,173], along with their documented toxicity on freshwater mussels, it is important to review the current regulatory guidelines defining the use of these chemicals. The U.S. passed the Microbead-Free Waters Act banning the manufacture of microbeads in 2017 (https://en.wikipedia.org/wiki/Microbead-Free_Waters_Act_2015 (accessed on 5 July 2023)) and banned the sale of cosmetic products containing microbeads in 2018 in an effort to control the spread of microplastics in the environment [172].

Currently, aquatic life criteria for CECs in general are not available at the national level. The USEPA issued a white paper to provide guidance to develop aquatic life criteria for CECs in 2008 [193], as "these chemicals have features that require additional consideration when applying existing ambient water quality criteria for the protection of aquatic life," using the USEPA's 1985 Guidelines for Deriving Numerical National Water Quality Criteria for the Protection of Aquatic Life and Their Uses' (https://www.epa.gov/wqc/contaminants-emerging-concern-including-pharmaceuticals-and-personal-care-products (accessed on 5 July 2023)). The white paper includes CEC toxicity information on many aquatic species, although it excludes freshwater mussels. Revised national aquatic life criteria on CECs are not available yet. Criteria exist for a few individual pollutants, such as the herbicide atrazine (draft criteria—Acute value: 3021 µg/L; Chronic value: 10 µg/L) and the petroleum derivative nonylphenol (Acute value: 55.5 µg/L; Chronic value 6.6 µg/L) [194,195]. However, these criteria do not incorporate toxicity information on freshwater mussels either. Canada has developed aquatic life criteria for more CECs; for example, Canada's chronic criterion of 10 µg/L on carbamazepine (CBZ) (https://ccme.ca/en/res/carbamazepine-en-canadian-water-quality-guidelines-for-the-protection-of-aquatic-life.pdf (accessed on 5 July 2023)) [196], though these criteria also do not incorporate impacts on freshwater mussels. It is noteworthy that CBZ can transform into other chemical forms in the aquatic environment, which have different effects on aquatic life. For example, CBZ-DiOH, the main transformation product of CBZ in Canadian waters, is biologically inactive [197] unlike, its counterpart in Minnesota waters, 10-OH-CBZ [198].

At the state level, a few states such as Michigan have developed drinking water standards for both surface and groundwater for PFOS (https://en.wikipedia.org/wiki/Perfluorooctanesulfonic_acid (accessed on 5 July 2023)) and PFOA (https://www.cancer.org/cancer/cancer-causes/teflon-and-perfluorooctanoic-acid-pfoa.html (accessed on 5 July 2023)) based on human health (http://connect.michbar.org/envlaw/blogs/environmental-law-journal/2018/05/15/the-abcs-of-emerging-contaminants (accessed on 5

July 2023)) and aquatic life criteria for a few pollutants. Minnesota has water quality standards for a handful of CECs, such as acetochlor (Chronic Standard (CS): 3.6 µg/L) and chlorpyrifos (CS: 0.041 µg/L) (https://www.revisor.mn.gov/rules/7050.0222/ (accessed on 5 July 2023)), but it has very few based on aquatic life. The Minnesota Department of Health (MDH)'s Drinking Water CEC initiative (http://www.health.state.mn.us/cec (accessed on 5 July 2023)) is one of the few frameworks available globally for monitoring and developing criteria for CECs in drinking water and groundwater based on human health [199]. Health-based values (HBVs) (https://www.health.state.mn.us/communities/environment/risk/guidance/hbvraawater.html (accessed on 5 July 2023)), which are concentrations of chemicals in drinking water that pose little or no risk to human health, are available for a number of CECs, such as a chronic standard of 3 µg/L (MPCA CS: 3.4 µg/L based on human health for Class 2A waters for atrazine (http://www.health.state.mn.us/divs/eh/risk/guidance/gw/table.html (accessed on 5 July 2023)), which based on available scientific evidence is protective of freshwater mussels in the long-term sense [87]. Water quality standards based on human health are intended for protection against accidental ingestion, and through consumption of aquatic life such as fish. These standards are not based on freshwater mussel toxicity information.

Having water quality standards for CECs combined with the MDH framework for CECs, is helpful in monitoring risks to human health and the environment, and therefore, should pave the way for more standards based on aquatic life in the near future. However, in the context of protecting beneficial uses, including aquatic life and human health, it is important to note three points. First, concentrations of CECs in waterbodies much lower than HBVs may bioaccumulate in fish and in sensitive species such as freshwater mussels [200,201], leading to detrimental effects on growth and survival [202,203] as well as potential future effects on human health [204]. Second, exposure to CECs is frequently accompanied by exposure to other environmental stressors, leading to multiple stressors that can vary with time, leading to adverse effects on the food web that could manifest across populations and through generations [205]. Finally, in large freshwater systems such as the Great Lakes [181] and their tributaries [206], these chemicals have been detected at significantly higher concentrations (with several sites showing exceedances by a factor of four or more) compared even to existing water quality and human health benchmarks compiled from a variety of sources [193,207–210]. This underscores the importance of developing aquatic life criteria for CECs and regulating point sources with corresponding WQBELs.

### 3.3.7. Wastewater Sensitivity

Mussels are sensitive to sewage and treated wastewater effluent, and can accumulate toxins such as CECs from these sources [211]. CECs and endocrine-disrupting chemicals (EDCs) are known to affect the immune and reproductive systems of aquatic animals, including freshwater mussels [212], and are documented to be widespread in Minnesota surface water, groundwater, and drinking water, especially near WWTPs, with signs of adverse aquatic impacts being documented [179,213]. As noted before, mussels are highly sensitive to CECs, with tolerance values for certain tested chemicals such as atrazine and glyphosate being 3.8 mg/L (chronic) and 0.5 mg/L (acute) [87,88]. Freshwater mussel consumption was historically prevalent among Native American tribes in North America [43], and indigenous peoples in other countries, such as Aborigines in Australia [214] and certain tribes continue the tradition [47]. However, as the consumption of mussels contaminated by wastewater effluent, metals, and radionuclides from mine waste can cause health problems in humans [41,215,216], the cost of treating these ailments can point to the benefits of regulating WWTPs and following stringent control measures in active or prior mine sites. Therefore, these costs should be factored into the benefits of upgrading/reclaiming such facilities/sites.

In summary, freshwater mussels are strongly linked with water quality and are affected adversely by a variety of water pollutants, with generally lower tolerance values compared to other aquatic species. Mussels have a complex relationship with nutrients (TN and TP),

which is detrimental to mussel survival at high nutrient and sediment loads. Mussels are especially sensitive to unionized $NH_3$, and are sensitive to major ions, heavy metals, and emerging chemicals. These effects may be worsened at high pH, and in low-flow conditions such as in winter months and dry summer months, in lakes and slow-moving streams, and in the presence of low hardness. Table 9 summarizes mussel tolerance values for selected pollutants from the toxicological literature discussed in this section.

**Table 9.** Mussel tolerance values for key pollutants.

| Pollutant | Mussel Tolerance Value | Source |
|---|---|---|
| Ammonia | 1.97–17 mg of TAN/L | USEPA 2013 [113] |
| Chloride | 113–1430 mg/L | Gillis 2011 [89] |
| Phosphorus | 0.05 mg/L | Morris et al., 2008 [93] |
| Nitrate and nitrite | 3 mg/L | Morris et al., 2008 [93] |
| Dissolved oxygen | >2–6 mg/L | Chen et al., 2001 [127] |
| CECs-Atrazine | 0.003 mg/L | Bringolf et al., 2007 [86–88] |
| Copper | 0.002–0.012 mg/L | Jorge et al., 2013 [91] |

## 4. Discussion: The Role of Regulation in Conserving Mussels

Based on our review and analysis of the literature in Section 3, freshwater mussels are sensitive to a wide variety of pollutants. However, nutrients, i.e., TN, TP, and related compounds such as $NH_3$, are different from other pollutants based on the multiple links connecting nutrients, phytoplankton, mussels, and species communities. It has been documented that phosphorus is the nutrient of greatest concern in Minnesota [217]. Based on our analysis of the water quality requirements of mussels, current WQS in Minnesota for several key pollutants, including nutrients and major ions such as chloride, may not be sufficiently protective of mussels. This inference suggests that prioritizing and managing pollutant loads could be key to conserving both mussels and the ecosystems that depend on them in order to ensure that further water quality regulation can play a reasonable role in ensuring effective protection for freshwater mussels. As a number of these pollutants, such as chloride, can accumulate in waterbodies and degrade habitats, a more comprehensive strategy that includes water quality regulations as well as other conservation strategies, such as improved wastewater treatment technology and best management practices (BMPs), could be more effective from a longer-term perspective.

### 4.1. Process to Develop Water Quality Standards

The AWQC developed by the USEPA use methodologies that are used by states when setting their own numeric values to protect beneficial uses of surface waters [218]. These beneficial uses include the protection of surface water quality as a source of drinking water, use by fish, macroinvertebrates and other aquatic organisms, recreation uses, and other uses.

The USEPA methodology uses results from published toxicity tests to calculate a numeric value that protects the aquatic community. Freshwater mussels are important organisms within the aquatic community and are often quite sensitive to contaminants in surface water. Ammonia is one example of an aquatic contaminant that mussels are sensitive to. As noted before, the USEPA published a revised AWQC for ammonia that better addresses the sensitivity of mussels and larval fish in 2013 [113]. Results from recent toxicity tests with early-life-stage mussels were critical to this revision. As is the case for many other aquatic organisms, the life stage of mussels exhibits a range of sensitivities, with the early life stage, called glochidia, often being the most sensitive. As stated earlier, Minnesota is preparing to begin rulemaking to adopt the 2013 USEPA AWQC for ammonia to replace its existing water quality standard for ammonia.

Minnesota follows the same procedures and methodologies used by the USEPA when developing WQS not available as AWQC. An example is the work on developing a WQS to protect aquatic organisms from nitrate exposure. Another example of Minnesota aquatic life WQS are those adopted for eutrophication in both rivers and lakes. Called "nutrient standards", these numeric values protect aquatic life from water quality conditions that would be harmful to aquatic organisms. In this case, phosphorus is the causative agent that leads to the growth of excessive algae, which degrades water quality.

### 4.2. Analyzing Sources of Water Pollution

Regarding pollution sources, based on Figure 1 in Section 3 both point and non-point sources share responsibility for mussel decline and extirpation in Minnesota. Figure 5 below combines Figure 1 with wastewater treatment outfalls to provide further evidence on this point. While non-point sources, mainly agricultural basins, align with high extirpation rates, areas with higher human density and a higher concentration of point sources such as WWTPs, coincide with mussel decline. There are fewer point sources in the north, where mussels are more protected, and more point sources in the south, where they are more at risk.

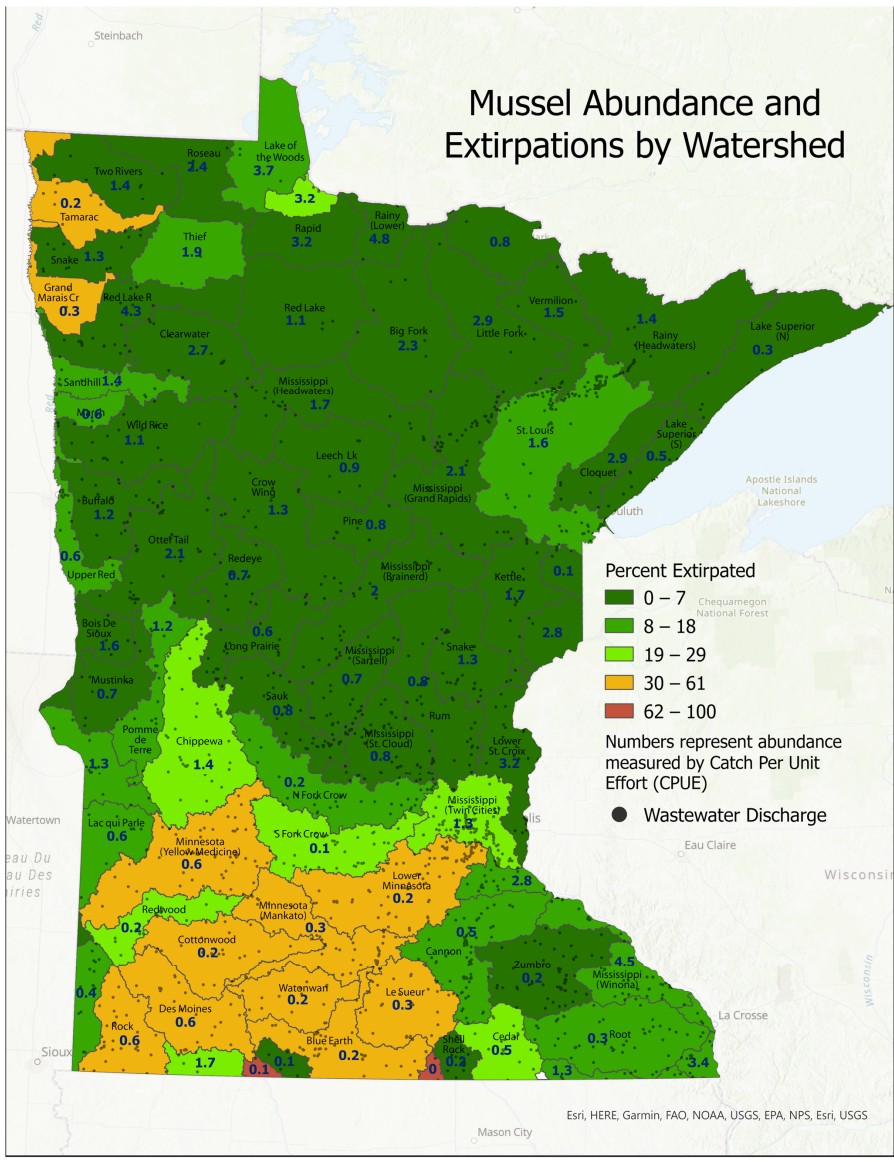

**Figure 5.** Point and non-point sources as probable causes of mussel decline.

Water quality regulation based on the Clean Water Act can currently be used to control point sources only. Pollution from non-point sources such as agricultural runoff and road salt is instead managed using a range of BMPs. However, point and non-point sources do not affect water quality uniformly. Based on watershed-level pollution data, non-point sources generally have a larger contribution to pollution than point sources. Figure 6 shows the contributions of both point and non-point sources for TP loads entering Lake Pepin over 20 years. Lake Pepin is a naturally occurring lake formed by the widest part of the Mississippi River. It receives drainage from half of the state's land area, including the upper Mississippi, Minnesota, and St. Croix rivers. Based on Figure 6, the total TP load entering Lake Pepin for all years of data availability (blue bars) is substantially higher than the total load from WWTPs (black line), among which the Metropolitan Plant in Saint Paul is a major contributor (dark orange line). This means that non-point sources are the major contributor of TP loads in Lake Pepin, and unlike point sources, there is no declining trend in their contribution.

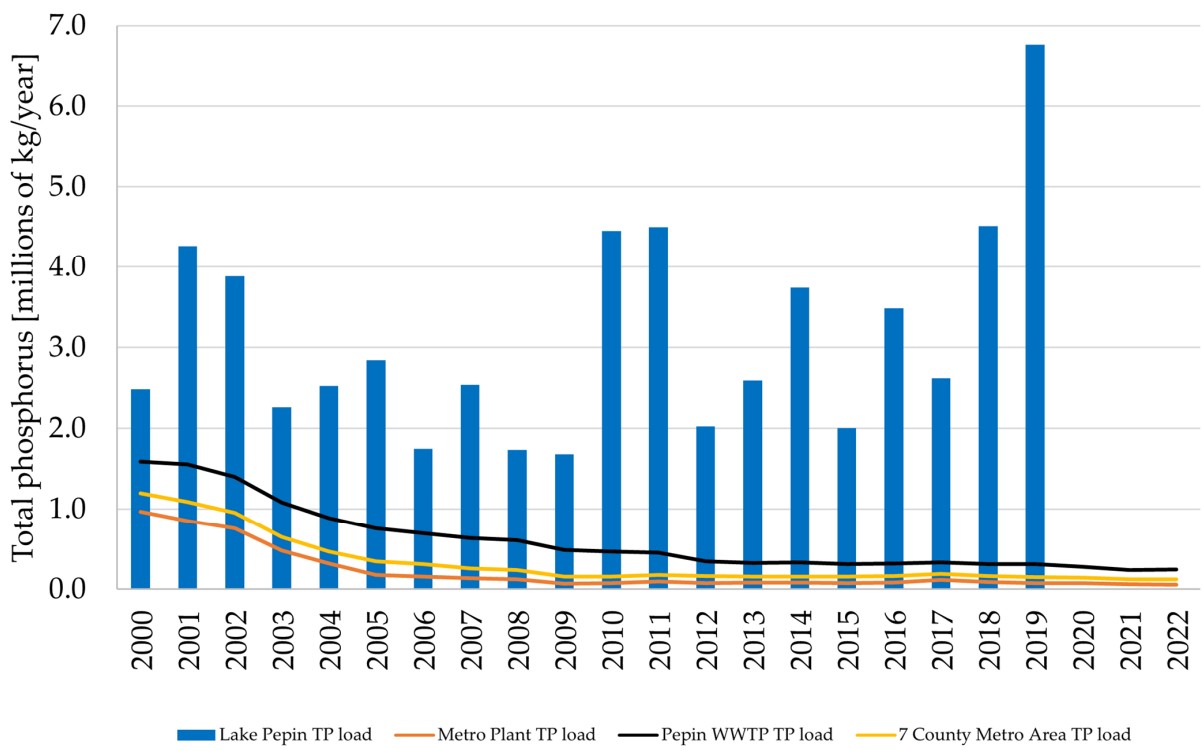

**Figure 6.** Sources and loads of phosphorus in Lake Pepin.

However, point sources contribute to pollution as well, and can seriously affect sensitive species such as freshwater mussels depending upon natural conditions such as the season, temperature, streamflow, and type of pollutant. Figure 7 represents a graph for a segment of the Minnesota River, showing point source TP contributions that can make up more than 60% of the total TP load during low-flow conditions in the summer months of June to September.

The possibility of relatively high nutrient loading from point sources depending on environmental conditions, is noteworthy because non-point sources are the major contributors to nutrient loading to the Minnesota River on average as its basin is dominated by croplands. Because point sources account for most of the nutrient load during low-flow periods, they contribute to eutrophication in Minnesota waters. During low-flow periods they can be a significant source of other pollutants as well, such as ammonia, chloride, sulfates, and emerging chemicals.

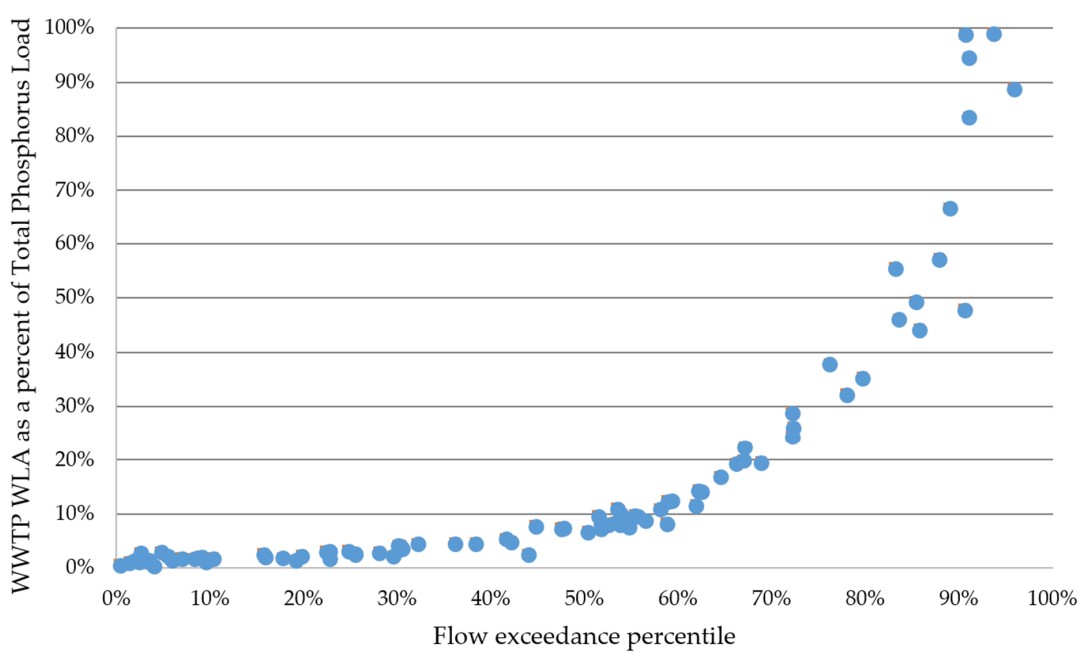

**Figure 7.** WLA as a percent of TP load for a Minnesota River reach under varying flow conditions.

An example of the impact of point source discharges on mussels is provided by a recent study of the Grand River in Ontario, a part of the Great Lakes watershed, which revealed a four-mile dead zone downstream of a WWTP where mussels historically had been abundant [219]. Mussels declined by 98% immediately downstream of the WWTP and by 60% downstream of the urban area served by the WWTP. Mussels found downstream featured larger size classes, indicating population effects on smaller mussels. The water near the plant had high levels of ammonia ($0.28 \pm 0.08$ mg/L) similar to USEPA 2009 criteria (the chronic standard for ammonia per the EPA 2009 draft AWQC was 0.26 mg/L at pH = 8, T = 25 °C). Average DO levels near the plant were 0.12 mg/L in 2012 but were found to be extremely low (0.08–1.3 mg/L) in 2006-2007 [220,221]. Phosphorus levels near the plant were 0.11 mg/L, while nitrate levels were more than twice the Canadian water quality guidelines in 2012 and nitrite levels >10 times than the guidelines ([222], page 16, Table 3). Available at Other nearby segments of the river downstream of the WWTP had high ammonia, as well as high nitrates ($NO_3^-$), P, chloride, and metals (http://greatlakesecho.org/2017/05/01/the-dead-zone-how-our-wastewater-is-killing-mussels/ (accessed on 5 July 2023)). The study concluded that eutrophication and loss of mussels resulted mainly from effluent high in ammonia and nitrites combined with nutrient cycling of N and P, which indirectly reduced DO levels. The plant underwent upgrades after 2012 that resulted in immediate water quality improvements. These results strengthen the links between population growth, urbanization, effluent discharges, and reduction in water quality [103], highlighting the need to regulate point sources for effective mussel conservation and sustainable ecosystem services flows.

While pollution from point sources can affect sensitive species such as mussels, point sources in general, have made more progress towards meeting state water quality goals compared to non-point sources. The clear declining trend in the TP load entering Lake Pepin from point sources, shown in Figure 6, is a reflection of a remarkable success story about point sources meeting phosphorus TMDL goals through sustained reductions over time; this was made possible by a combination of strategies, including the MPCA Phosphorus reduction strategy in 2000 (https://www.pca.state.mn.us/water/phosphorus-wastewater (accessed on 5 July 2023)), state funding for improved wastewater treatment, regulation targeting eutrophication in lakes and rivers, i.e.,LES and RES [130,131], a systematic process of monitoring and reporting of pollutant data, and enforcement of water quality standards. This overall picture for sustained reduction in point source TP contributions is supported

by Figure 8, which breaks the contributions down in terms of municipal and industrial loads.

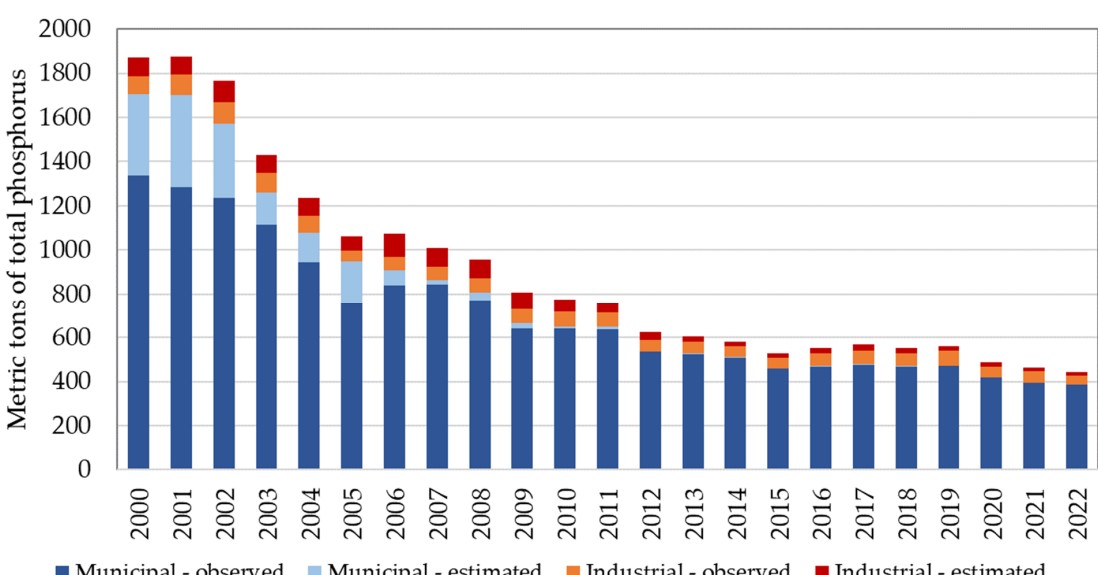

**Figure 8.** TP loads from municipal and industrial WWTPs.

Point sources are not major contributors to nutrient loading in an aggregate sense of the term. In addition, point sources such as municipal WWTPs have made remarkable progress in meeting nutrient TMDL goals, particularly for TP, so that additional regulation could be both costly [223] and insufficient to protect mussels. As indicated by Figure 8, such regulation may not provide sufficient return on investment. In order to use water quality regulation effectively for mussel conservation, it must be targeted to generate maximum pollution reduction at a reasonable cost to society. Based on the current WQS for nutrients, there could be two potential ways to revise the WQS to protect mussels from nutrient pollution: first, adopting new WQS or revising existing WQS incorporating freshwater mussel data (for example, Minnesota's planned adoption of the USEPA's current ammonia criteria would be helpful to mussels), second, as Minnesota does not currently have a nitrate standard, adopting a new nitrate standard that considers the toxicity of nitrate to freshwater mussels could be helpful.

Finally, as freshwater mussels are affected in multiple ways by phosphorus, a revision of the RES criteria accounting for mussels could be an additional step in protecting them from nutrient pollution. While Minnesota's RES criteria are not directly based on mussels, they are significant for three reasons. Minnesota currently does not have stand-alone aquatic life standards for nitrate or phosphorus, so that having the RES criteria helps bridge the gap in water quality protection for valuable beneficial uses such as aquatic life use and recreation. In addition, eutrophication is a critical link in the relationship between mussels and nutrients such as nitrates and phosphorus. Excess nutrient loading in streams causes eutrophication, reducing DO levels and triggering $NH_3$ discharge by toxic algal blooms, both of which are detrimental to mussel growth and survival. By controlling TP and corresponding eutrophication through algal blooms, RES could contribute to controlling the formation of unionized $NH_3$ in waterbodies. Finally, the RES for TP is above 35 µg/L, above which streams are considered eutrophic in Canada. Therefore, updating the RES to account for mussel survival in the absence of separate aquatic life standards for nutrients would be a significant step in protecting mussels in Minnesota.

## 5. Future Directions: Determining an Effective Mussel Conservation Strategy

Freshwater mussels are affected by a variety of pollutants, including nutrients, major ions such as chlorides and sulfates, and CECs. Current Minnesota water quality standards

(WQS) for several of these pollutants, for example unionized $NH_3$, TP, and Cl, are not protective of mussels. Based on the measurable beneficial effect of laws such as the Clean Water Act on water quality in Minnesota over time, it is plausible that regulating pollutants that are harmful to sensitive species such as freshwater mussels could be effective in conserving them by protecting existing species and aiding in the recovery of critically endangered species.

This beneficial effect of regulation has been documented for *Hexagenia* mayflies (*Hexagenia limbate*) which, like freshwater mussels, are important to the aquatic food web and sensitive to water pollution, and as such are considered biomarkers of water quality. For example, changes in the distribution of *Hexagenia* mayflies were used to assess water quality in the Upper Mississippi River during the periods 1957–1976 and 1985–1986 by [224] Based on sampling data in 1957–1976, mayflies were abundant in most of the 29 navigational pools in the Upper Mississippi River and almost non-existent in Pool 2 (the reach downstream of the Minneapolis–Saint Paul metropolitan area) and Lake Pepin. The absence of mayflies in Pool 2 was mainly due to pollution from the Metropolitan WWTP (the Metro Plant, which treated 81% of the sewage from the metro area), leading to very low DO concentrations and high levels of TSS. In 1985–1986, however, mayflies were found to be abundant in Pool 2 and Lake Pepin to the point of being a nuisance, indicating a dramatic improvement in water quality as measured by a reduction in TSS and an increase in DO in these waterbodies. This was over ten years after the introduction of the Clean Water Act, a new WQS for DO at 5 mg/L, and mandatory secondary treatment of wastewater and pollution abatement from the Metro Plant and other point sources [225]. These findings highlight two points, namely, that pollution can severely impact sensitive species, and that regulating pollution can significantly improve water quality and aid in the recovery of such species.

Owing to the severity of the current imperiled status of freshwater mussels, their sensitivity to a variety of pollutants, and the complex pathways between mussels, pollutants, and ecosystems, it is not clear that regulating any single pollutant would be effective in conserving mussels. Second, while both point and non-point sources may contribute towards pollutants that affect mussels, new WQS would mean additional mandatory reductions from point sources through incentivized BMPs, cooperative management efforts, and other non-regulatory abatement strategies, although non-point sources may continue to contribute. Finally, realized reductions from point sources could be limited for several reasons, for example due to diminished returns in obtaining additional abatement for certain pollutants such as TP (for which point sources have already achieved remarkable success), extremely high costs for abatement technology for pollutants such as chloride and TP, and general affordability issues for wastewater treatment infrastructure and technologies for small communities in Minnesota.

For these reasons, a strategy for effective mussel conservation needs to satisfy a combination of linked criteria: (1). an *ecological* criterion that identifies a set of pollutants for which regulation will be the most beneficial to mussels, which may include a subset of key mussel habitats where reducing discharges of the above pollutants would lead to maximum benefit; and (2). an *economic* criterion that identifies a group of pollutants from the above subset for which any additional mandatory reductions on point sources can be justified based on the 'polluter pays principle' (https://en.wikipedia.org/wiki/Polluter_pays_principle (accessed on 5 July 2023)) and where further abatement could be achieved at a reasonable cost. This crietrion may include a set of point sources with the highest potential for further abatement. These criteria are described below.

**Ecological criterion**. Freshwater mussels are sensitive to a variety of pollutants, with effects varying across species, age classes, habitats, and interactions between these factors. Exogenous factors such as land use changes, human activity, natural disasters, and climate change influence these relationships as well. This means that choosing a specific pollutant that mussels are sensitive to for a new or modified WQS may not actually benefit mussels. For example, as freshwater mussels are affected by excessive nutrient loading, regulating nutrient pollution such as TP could be key to mussel conservation. However, controlling

TP may not be as effective if most of the phosphorus load in the relevant waterbody is linked to sediment [226], such as in Lake Pepin, or if phosphorus is not the limiting nutrient in that waterbody (https://cms5.revize.com/revize/columbiaheights/document_center /Stormwater/Phosphorus_201412041423196128.pdf (accessed on 5 July 2023)). Similarly, controlling chloride in waterbodies close to WWTPs could make more of a contribution to protecting mussels than in waterbodies far away from WWTPs and urban regions [17]. Recent research in the Saint Croix River has shown that declines in mussel abundance and diversity could happen even in the most pristine and protected habitats without any significant human impacts, and could vary between groups of species based on biological assemblages, community structure, and life history traits, highlighting the severity of freshwater mussel imperilment and the need for careful and effective planning in less pristine river systems [21].

Therefore, an ecological criterion accounting for these pathways and interactions is needed in order to determine the set of key pollutants and mussel habitat combinations that would be most responsive to regulatory or other conservation measures. Ideally, the ecological criterion should be based on inferences gathered from models examining relationships between mussel and water quality metrics. These relationships could be further examined by changing exogenous parameters such as flow regimes and habitats. By aggregating inferences from all such findings, a key set of pollutants for a matched set of mussel habitats in which a targeted approach would be beneficial can be determined.

Based on this information, it would be helpful to investigate the nature and location of mussel habitats in combination with a range of factors, including proximity to point source discharge locations for key pollutants, non-point sources such as croplands, and urban centers. Such an investigation is the first step in developing ecological criteria delineating those mussel habitats in Minnesota where pollution reduction is likely to have the highest conservation potential.

**Economic criterion.** Regulation as a way of conserving mussels implies additional regulation of point sources. Therefore, the economic criterion needs to ensure that these additional regulations are effective in conservation while being achievable at a reasonable cost to society. To ensure that these additional regulations to protect imperiled species are fair and effective, the polluter-pays principle needs to be met for corresponding point source and pollutant combinations, a set of mussel habitats close to point sources should benefit from such regulations, and any additional costs for such abatement should be reasonable.

Water quality monitoring data show that while point source contributions to nutrient loading represent only a small fraction of total nutrient loads, they can dominate nutrient loading for specific pollutants such as TP during low-flow periods in the summer season, leading to eutrophication, which can affect freshwater mussel survival due to lack of DO as well as production of unionized $NH_3$ from toxic cyanobacteria. Therefore, nutrient loading from point sources could be the main reason for mussel declines during the summer months, provided the pollutants are not dissipated and are held in the waterbody for most of the season, thereby having an impact on the corresponding ecosystem. These conditions are most common in slow-moving urban rivers and shallow lakes during summer and can be reinforced by exogenous events such as droughts and land use changes. Point sources can dominate in terms of the relative contribution during low-flow periods for major ions such as chloride as well. Current research has indicated that WWTPs are a significant source of chloride pollution owing to Minnesota being a hard water state and consequent high use of home water softeners [17]. While the contribution of point sources remains the same year-round, the relative contribution is higher in summer owing to less runoff from non-point sources such as road salt use. In this case, urban regions such as the Twin Cities metropolitan area have a lower relative contribution from point sources compared to rural regions with higher demand for home water softening and lower road salt use. Finally, as noted in a study on the Grand River in Ontario, freshwater mussels are sensitive to wastewater effluent.

Based on the above examples, a suitable blend of mandatory regulation of point sources for specific pollutants during summer months and incentivized BMPs for non-point sources such as cover crops and buffers during average to high flow periods for key waterbodies with known mussel habitats could satisfy the polluter-pays principle. The first step in applying the economic criterion to mussel conservation in Minnesota is to examine whether the polluter pays principle can be applied to a set of mussel habitats and point source combinations to benefit mussels. This would entail a spatial study of the location of large versus small dischargers of key pollutants as well as non-point source locations upstream of the mussel habitats studied in the ecological criterion.

It is important to note that even if there exists an optimal set of pollutants key to mussel conservation and a set of key mussel habitats where these pollutants should be reduced, questions may arise as to whether the economic criterion is met, i.e., whether further reductions in these pollutants could be achieved at a reasonable cost, and if so whether these reductions could be achieved at any point source or would be best achieved by targeting a subset of point sources and mussel habitat combinations where both ecological and economic aims could be met. This would mean targeting a subset of key mussel habitats and corresponding point sources that are capable of reducing discharge levels in order to protect mussels at a reasonable cost.

Based on these two key criteria, in order to determine whether regulating point source nutrient discharges would be effective in conserving mussels it is necessary to examine the following points: (1) whether point sources such as municipal or industrial WWTPs are significant contributors to nutrient loading, particularly during low-flow conditions; (2) whether past regulation of point sources has averted further declines in water quality and aquatic life; and finally (3) whether there is a specific combination of nutrients for which regulation would be ecologically effective for a set of mussel habitats and a corresponding set of point sources that would be able to achieve the required reduction in discharges at a reasonable cost.

Based on the results of the series of studies above, the next step could be to investigate potential dischargers for a range of regulatory actions as well as BMPs for non-point source reductions at a reasonable cost to stakeholders, including both the regulated parties and water users. In the final step, the ecological and economic criteria can then be combined to produce a range of possible scenarios with conservation potential, along with their costs and benefits, selecting those that provide the highest return on investment. Such an approach would represent an effective conservation strategy for freshwater mussels in Minnesota.

## 6. Conclusions

In this paper, we have explored the conservation needs of freshwater mussels given their declining status and valuable contribution to ecosystem services which underpin various beneficial uses of clean water. As the goal of protecting ecosystem services is implicit in the Clean Water Act, we argue that protecting mussels is a cost-effective way of achieving clean water goals. With the goal of identifying gaps between the water quality needs of mussels in terms of pollutant levels and the current water quality standards in Minnesota, we reviewed and analyzed the literature in three topics: the contribution of mussels to ecosystem services, their links with water quality, and the threats they face from water pollution and protection available from current regulations.

We found effective mussel conservation to be a complex problem that needs to account for multiple factors, including their declining status, multiple threats to their conservation, sensitivity to a variety of pollutants, gaps in current science and policy, and the costs and benefits of additional regulation to multiple stakeholders. Within the context of the Clean Water Act, we have discussed the role that water quality regulation can play in conserving mussels. Finally, as a future direction in freshwater mussel conservation, based on the results of our review we have proposed a framework for an effective conservation strategy for mussels in Minnesota using a blend of regulatory and non-regulatory best management

practices based on both ecological and economic criteria. These criteria should ensure the maximum return on conservation investment in mussels at a reasonable cost to society. We hope that the information in this paper will be helpful to future research on freshwater mussels and benefit more cost-effective decision-making on clean water goals by adding the lens of ecosystem service protection through mussel conservation.

**Author Contributions:** Conceptualization, B.B.; methodology, B.B.; validation, All; formal analysis, B.B.; investigation, B.B., R.W.B.J., and D.H.; resources, all; data curation, B.S.; writing—original draft preparation, B.B., R.D., P.M., and D.W.; writing—review and editing, All; visualization, B.B., R.W.B.J., R.D., D.H., and D.W.; supervision, B.B., R.W.B.J., D.H., and B.S.; project administration, B.B. All authors have read and agreed to the published version of the manuscript.

**Funding:** This research received no external funding.

**Data Availability Statement:** The data presented in this study are not publicly available due to being part of a rare resources database that is covered by State privacy restrictions.

**Acknowledgments:** We are grateful to William Cole for his review, edits, comments, and support, which improved the quality of the paper. We gratefully acknowledge Casey Scott for his helpful review and comments, and for providing Figures 6 and 8. We gratefully acknowledge Zebulin Secrist for providing updated data on MDNR mussel surveys, and David Strayer for his permission to use Figure 4.

**Conflicts of Interest:** The authors declare no conflict of interest.

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
