# Peer review of "Freshwater Mussels, Ecosystem Services, and Clean Water Regulation in Minnesota: Formulating an Effective Conservation Strategy"

_water, doi:10.3390/w15142560_

Round 1

Reviewer 1 Report

The manuscript can be published in the present form.

Author Response

Dear Reviewer,

Thank you for your helpful review of our manuscript. We are happy to see that you find our manuscript a significant contribution to the field. We are also happy to know that you find our manuscript organized and comprehensively described, scientifically sound, and with adequate references. We note your statement about the English needing improvement in terms of being correct and readable and we will address these issues in our revision. 

Thank you again.

Baishali Bakshi on behalf of author team.

Reviewer 2 Report

The manuscript " Freshwater mussels, ecosystem services, and clean water regulation in Minnesota: Formulating an effective conservation strategy" reports 8% of freshwater mussels have been lost and 60% of remaining species under threat from a variety of factors in Minnesota. The author believes that water quality is the most critical factor affecting the survival of mussels and finds that Minnesota's current water quality regulations are insufficient to protect freshwater mussel. Therefore, the author proposes this policy of protecting mussels “preserving and improving the quality of public water resources to sustain the ecosystem services they provide for human well-being and aquatic life uses”.

At the same time, the author found that the loading of nutrients, as well as salts, metals, and emerging chemicals such as chlorides and sulfates, is the main threat to Minnesota's water quality. In response to these threat factors, the author has developed a strategic framework for effective protection of mussels based on ecological and economic standards, which can provide more cost-effective decisions to achieve the goal of clean water from the perspective of protecting ecosystem services for mussels.

By analyzing the contribution of freshwater mussels to the ecosystem, the harm of water pollutants to mussels and the effect of water quality regulation on mussels, the author summarizes the importance of protecting mussels and the main pollution sources that harm mussels. In addition, critical analyses of past findings and current water quality standards, Proposed specific pollutant limitations in the Clean Water Act.

Overall, the results are interesting and determined an effective mussel conservation strategy. However, further revisions are necessary before this manuscript can be accepted for publication in water.

Specific comments are listed as follows:

1. The whole article describes in detail the contribution of mussels to maintaining the balance of the ecosystem and the damage of different pollutants to it. However, the authors suggest that an effective mussel conservation strategy that satisfies both "ecological and economic criteria" is a difficult strategy to implement.

2. Line 29: Delete this sentence “Incorporating mussel conservation into wat”.

3. Line 36: [Cope et al., 2021; Haag et al., 2019; Haag 35 and Williams, 2014, Williams et al. 1993], Separate literature with semicolons.

4. Line 126: Does this belong to the content of materials and methods?

5. Line 146-155: This part has been repeatedly mentioned in the article, and there is no need to describe it here, it is recommended to integrate it.

6. There is too much description in this section (3.1. Mussels and ecosystem services), it is recommended to simplify it.

7. The paper is written clearly and was not difficult to understand, but it would benefit from editing for improved English usage and grammar.

8. Overall, the submitted manuscript needs exhaustive proofreading.

Author Response

Thanks for your review and comments. We are happy to note that you find our results interesting and determined an effective strategy. We have provided responses to your comments below and also addressed your comments in the revised manuscript. 

Comment 1:

  1. The whole article describes in detail the contribution of mussels to maintaining the balance of the ecosystem and the damage of different pollutants to it. However, the authors suggest that an effective mussel conservation strategy that satisfies both "ecological and economic criteria" is a difficult strategy to implement.

Response: Mussels are imperiled and affected by a variety of pollutants from different sources. But all sources do not contribute similarly to pollution and also have different costs and benefits associated with regulation and treatment technology. This makes mussel conservation a complex problem, so that an effective strategy needs to account for both ecological and economic criteria. Implementation of such a strategy is feasible. However, we wanted to acknowledge it can be difficult owing to its complexity as well as the time and resources involved in regulating pollution.

2. Line 29: Delete this sentence “Incorporating mussel conservation into wat”. Response: We have deleted this sentence.

3. Line 36: [Cope et al., 2021; Haag et al., 2019; Haag 35 and Williams, 2014, Williams et al. 1993], Separate literature with semicolons.

Response: We have separated literature with semicolons.

4. Line 126: Does this belong to the content of materials and methods?

Response: The combined literature review from three topics and the critical analyses from past findings, is our materials and methods for this paper and is therefore described in the Materials and Methods section. We moved the statement “To our knowledge, this is the first review of combined and disparate literatures that collectively provide information on conservation as well as clean water goals.” , to the first paragraph in the Literature Review section (Section 3).

  1. Line 146-155: This part has been repeatedly mentioned in the article, and there is no need to describe it here, it is recommended to integrate it.

Response: We have integrated the two sub-sections into one sub-section: ‘Food provisioning, aquatic recreation, and cultural services’ and made it more compact.

  1. There is too much description in this section (3.1. Mussels and ecosystem services), it is recommended to simplify it.

Response: We have simplified the description in the revised manuscript.

  1. The paper is written clearly and was not difficult to understand, but it would benefit from editing for improved English usage and grammar.

Response: We have carefully edited the paper for improvement in English usage and grammar.

8. Overall, the submitted manuscript needs exhaustive proofreading.

Response: We have extensively proofread the revised manuscript.

Reviewer 3 Report

Freshwater mussels provide important ecosystem services, but the mussels are declining at an alarming rate. This ms is an interesting combination of review and recommendations for management and conservation strategy. I enjoyed reading the ms.

- Pelase correct Vaughn 2017 --> as Vaughn 2018 throughout the ms

The ms consisted of a review on ecosystem services of freshwater mussels and on relationship between freshwater mussels and water quality including pollutants, including identification of gaps in protection provided by current water regulation of Minnesota. A framework for determining an effective conservation strategy for mussels in Minnesota (based on ecological and economic criteria) was formulated. Because freshwater mussels are important ecosystem engineers and because they have become endangered, the topic is relevant for the field. It identified gaps in the current water regulation of Minnesota. It provides a conservation strategy to ensure adequate conservation of freshwater mussels at reasonable cost. In my view, this is a novel approach and should be useful also in other regions and countries. Yet here was no experimentation or sampling design requiring controls.

Author Response

Thank you for your review and comments on our manuscript. We are happy to note that you found our paper interesting and thought it was a novel approach that can be applied in other places. 

We did not have experimentation or sampling design in this paper as it is meant to present the problem and design the framework for a solution using a strategy based on multiple criteria. This paper is the first part of a multi-part project, and we will be designing and applying methodology and implement the strategy in later phases. 

We corrected the year for the Vaughn reference as 2018 throughout the paper.

Baishali Bakshi on behalf of the author team.